# The productivity-biodiversity relationship varies across diversity dimensions

Philipp Brun [1]*, Niklaus E. Zimmermann [1], Catherine H. Graham[1], Sébastien Lavergne[2], Loïc Pellissier[1,3], Tamara Münkemüller [2] & Wilfried Thuiller[2]

Understanding the processes that drive the dramatic changes in biodiversity along the productivity gradient remains a major challenge. Insight from simple, bivariate relationships so far has been limited. We combined >11,000 community plots in the French Alps with a molecular phylogeny and trait information for >1200 plant species to simultaneously investigate the relationships between all major biodiversity dimensions and satellite-sensed productivity. Using an approach that tests for differential effects of species dominance, species similarity and the interplay between phylogeny and traits, we demonstrate that unimodal productivity–biodiversity relationships only dominate for taxonomic diversity. In forests, trait and phylogenetic diversity typically increase with productivity, while in grasslands, relationships shift from unimodal to declining with greater land-use intensity. High productivity may increase trait/phylogenetic diversity in ecosystems with few external constraints (forests) by promoting complementary strategies, but under external constraints (managed grasslands) successful strategies are similar and thus the best competitors may be selected.

[1] Swiss Federal Research Institute (WSL), 8903 Birmensdorf, Switzerland. [2] Univ. Grenoble Alpes, CNRS, Univ. Savoie Mont Blanc, LECA, Laboratoire d'Écologie Alpine, F- 38000 Grenoble, France. [3] Landscape Ecology, Institute of Terrestrial Ecosystems, Department of Environmental Systems Science, ETH Zürich, 8092 Zürich, Switzerland. *email: philipp.brun@wsl.ch

Biological diversity changes dramatically along the gradient of ecosystem productivity. This is particularly visible in plant communities that transform from marginal grasslands high above the tree line to semi-natural meadows and pastures and finally to lush broad-leaved forests[1]. Understanding how ecological processes, such as environmental filtering, competitive exclusion, or evolutionary context affect these transformations remains an unresolved challenge in ecology (see ref. [2] for a review). Here, we follow a recent call to employ approaches that embrace more complexity[2] and show that simultaneously and systematically exploring productivity–biodiversity relationships across several biodiversity dimensions may provide deeper insight than focusing on one biodiversity dimension alone.

Two approaches have commonly been used to study the relationship between productivity and biodiversity. First, the shape and strength of productivity–biodiversity relationships have been studied intensively in natural and semi-natural systems based on observations. Observational studies mostly identified unimodal productivity–biodiversity relationships[3,4], but alternative relationships were also found (e.g., ref. [5]). Biodiversity of the typically large numbers of species considered in such studies has commonly been approximated by species richness. Second, manipulation experiments have been conducted to test to which extent biodiversity promotes productivity. Such experiments identified positive associations between several biodiversity dimensions and productivity[6–10]. However, these findings may not be directly comparable to observation-based assessments since their typically local extent leads to comparably narrow productivity ranges covered, and because biodiversity change in experiments is induced by artificial community modifications[11,12]. Observational studies, in essence, may thus offer a less distorted picture of productivity–biodiversity relationships in natural communities, but so far have been limited by the rather crude assumption that biodiversity equals species numbers.

Species richness ignores how communities are structured by both species dominance and similarity. Biodiversity estimates accounting for species dominance assign a higher weight to species with higher coverage or higher abundance in a community. With the same species richness, dominance-corrected biodiversity is higher for a community where abundances are more evenly distributed in comparison to a community dominated by a few common species. Additionally, species are not equal in terms of their functions and ecological strategies, and thus the similarity between species should be considered, with higher biodiversity in communities where species are more dissimilar[13]. Ecologically relevant similarity information may be obtained directly, by measuring functional traits, or indirectly, by comparing the positions of species on a phylogenetic tree. Biodiversity dimensions considering similarity information can be used to test ecological theories that link species similarity directly to their performance in abiotic (e.g. environmental filtering) and biotic environments (e.g. competitive exclusion).

A comprehensive analysis of biodiversity and its response to various ecological processes may be obtained by systematically varying the assumptions on species dominance and similarity in biodiversity estimates (Fig. 1a). Even when accounting for dominance and similarity, using only small sets of biodiversity metrics may deliver an incomplete picture of productivity–biodiversity relationships and will be constrained by the assumptions of the assessed metrics. This limitation can be relaxed by systematically investigating the realm of major assumptions. For example, the assumed importance of species dominance can be continuously increased in Hill's numbers framework[13,14] using the parameter $q$ (Fig. 1a). A lineage of similar species that dominates communities along an environmental gradient may then result in low beta diversity estimates above a certain value for $q$, but not when $q$ is set to zero and all species are treated as equally important[15].

Assumptions on species similarity may be systematically varied along two axes (Fig. 1a). The first axis is linked to the type of similarity information, with poles defined by trait similarity and phylogenetic relatedness[16]. Along the axis, trait and phylogenetic similarity information are then combined to functional-phylogenetic similarity estimates, using a weighting parameter, $a$, that balances their contributions[16]. On the second axis, species similarity assumptions can be varied by emphasizing high versus low similarities (i.e., the scale of similarity)[15,17,18]. In other words, branches close to the tips or branches close to the roots of the phylogenetic or functional tree may be emphasized (see Fig. 1a).

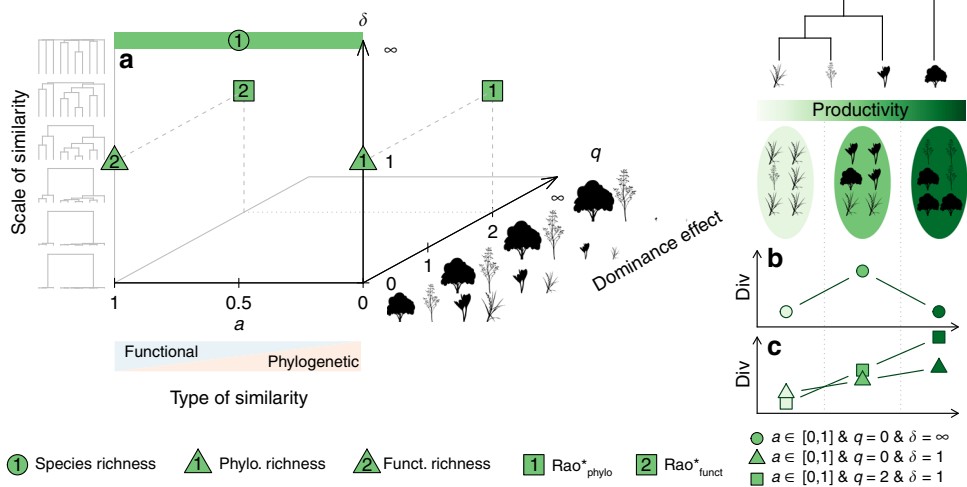

**Fig. 1 Definition of biodiversity space and hypothesized productivity–biodiversity relationships.** Panel (**a**) shows a sketch of our definition of biodiversity space. Functional-phylogenetic weighting parameter ($a$) on the $x$-axis corresponds to increasing emphasis on trait similarity relative to phylogenetic similarity. Scaling parameter $\delta$ on the $z$-axis represents increasing emphasis on small versus large species differences. Dominance weighting parameter ($q$) on $y$-axis represents increasing emphasis on dominant species in biodiversity estimates. Expected relationships between species richness and productivity is shown in panel (**b**); expected relationships between trait/phylogenetic diversity metrics and productivity are shown in panel (**c**). The green symbols in the diagrams represent commonly used biodiversity metrics (see legend). Rao* represents $1/(1-\text{Rao})$, with Rao being Rao's quadratic entropy[68]. Silhouette images are courtesy of Philipp Brun.

To this end Pagel's $\delta$-transformation[18] can be used, where higher values for $\delta$ give greater emphasis on high versus low similarity. In forest tree communities, biodiversity estimates considering low similarity scales (low $\delta$), focusing e.g. on different clades, respond to the environment, whereas estimates considering high-similarity-scales (high $\delta$), focusing e.g. on sister species, appear to be influenced by competition[17].

Considering the above metrics allows us to begin to tease apart different ecological hypotheses. The relationship between productivity and species richness, for example, may be unimodal, since according to the most prominent historical hypothesis[19–21], species numbers decline in unproductive environments due to environmental filtering, while in productive environments lower species richness results from increased competition and, in turn, competitive exclusion. In contrast, the limiting similarity hypothesis assumes species to specialize on complementary niches (herein the term niche is used in the Grinnellian sense[22]) in communities where competition is high, promoting assemblages with diverse traits[23–25]. Similarly, in such communities phylogenetic diversity was observed to be over-dispersed[26]. Trait and phylogenetic diversity therefore potentially increase with productivity (Fig. 1c). However, productivity–biodiversity relationships also depend on the local environment.

Both productivity and biodiversity are modulated by climate, ecosystem type, and land use intensity which likely also affects their interrelationship. Climate constrains both ecosystem productivity and the number of species capable to thrive. In harsh climates, the observed productivity gradient may be shifted to lower values and also the observed biodiversity may be lower than in more favorable climates. Moreover, biological communities differ across ecosystem types such as grasslands and forests, with potential implications on the shape of the productivity–biodiversity relationships. This is particularly true for semi-natural ecosystems which are under the influence of contrasting forms of land use. In semi-natural grasslands, for example, extensive management usually includes the removal of a significant fraction of the biomass above a certain height. This reduces light competition and filters for species capable of persisting despite the frequent cutting and/or grazing[19]. Conversely, increasing levels of fertilizing may increase productivity and thus competition within communities. In forests, on the other hand, plant height is rarely artificially constrained. Light competition may therefore be stronger in forests than in grasslands. However, forest communities may sometimes be artificially altered through plantations and promotions of desired tree species.

Here, we investigated productivity–biodiversity relationships for an extensive set of plant community observations in the French Alps (Fig. 2a), covering three climate zones (Fig. 2b) and two major ecosystem types at several levels of land use intensity. We combined over 11,000 diverse, geo-referenced plant community surveys mainly from forests and perennial grasslands, a genus-level phylogeny, and three key functional traits (specific leaf area (SLA), height and seed mass (SM)) for over 1200 species of vascular plants to comprehensively investigate productivity–biodiversity relationships. Our study area included over 40,000 km$^2$ in the lower montane, upper montane, and lower alpine bioclimatic zones[27]. Given this massive number of community plots, traditional in situ measurements of plant productivity were not feasible. We therefore estimated productivity with the high-resolution, satellite-sensed Normalized Difference Vegetation Index (NDVI) from the Landsat program (landsat.gsfc.nasa.gov) which has been demonstrated to be good substitute at regional scales[28]. For clarity, we focused on the common and complementary indices of species richness and Rao's quadratic entropies of unscaled functional and phylogenetic diversity (Rao*$_{funct}$ and Rao*$_{phylo}$,

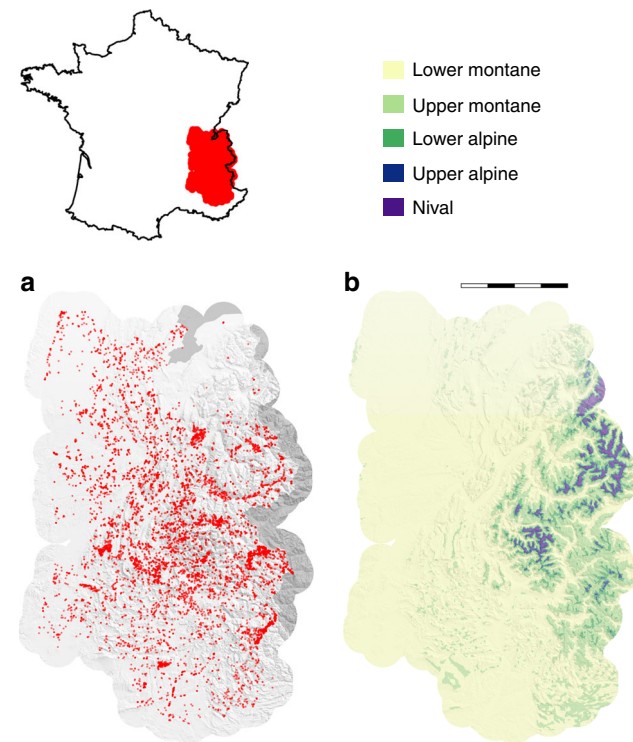

**Fig. 2 Distribution of community plots and bioclimatic zones.** Panel (**a**) shows the distribution of the community plots in the French Alps. Panel (**b**) indicates corresponding bioclimatic zones, as defined by ref. [27]. Total length of scale bar is 80 km. Source data are provided in the Source Data file.

respectively), but corresponding results across the whole biodiversity space introduced above (Fig. 1) are also provided, either in the Results section or in the Supplementary material.

Based on this set-up, we test and largely confirm the following hypotheses:

(1) We expect contrasting productivity–biodiversity relationships for different dimensions of biodiversity.

   a.
   We assume environmental filtering and competitive exclusion to shape unimodal relationships between species richness and productivity (Fig. 1b), and find such relationships in all ecosystem types.
   b.
   We assume niche differentiation in competitive environments to result in a monotonically increasing relationship between productivity and trait and phylogenetic diversity (Fig. 1c), and find such relationships across all land cover types and in forests, but not in grasslands, where unimodal relationships prevail.

(2) We expect differences between productivity–biodiversity relationships between bioclimatic zones, between ecosystem types, and under different land use intensities, and find them in particular for ecosystem types and land use intensities.

Productivity may increase trait/phylogenetic diversity in forests because, within the relatively large niche space available, competitive exclusion selects for species that pursue complementary ecological strategies. In (managed) grasslands niche space is externally constrained and successful ecological strategies are very similar. Competitive exclusion may thus select for the growth forms with the highest competitive ability.

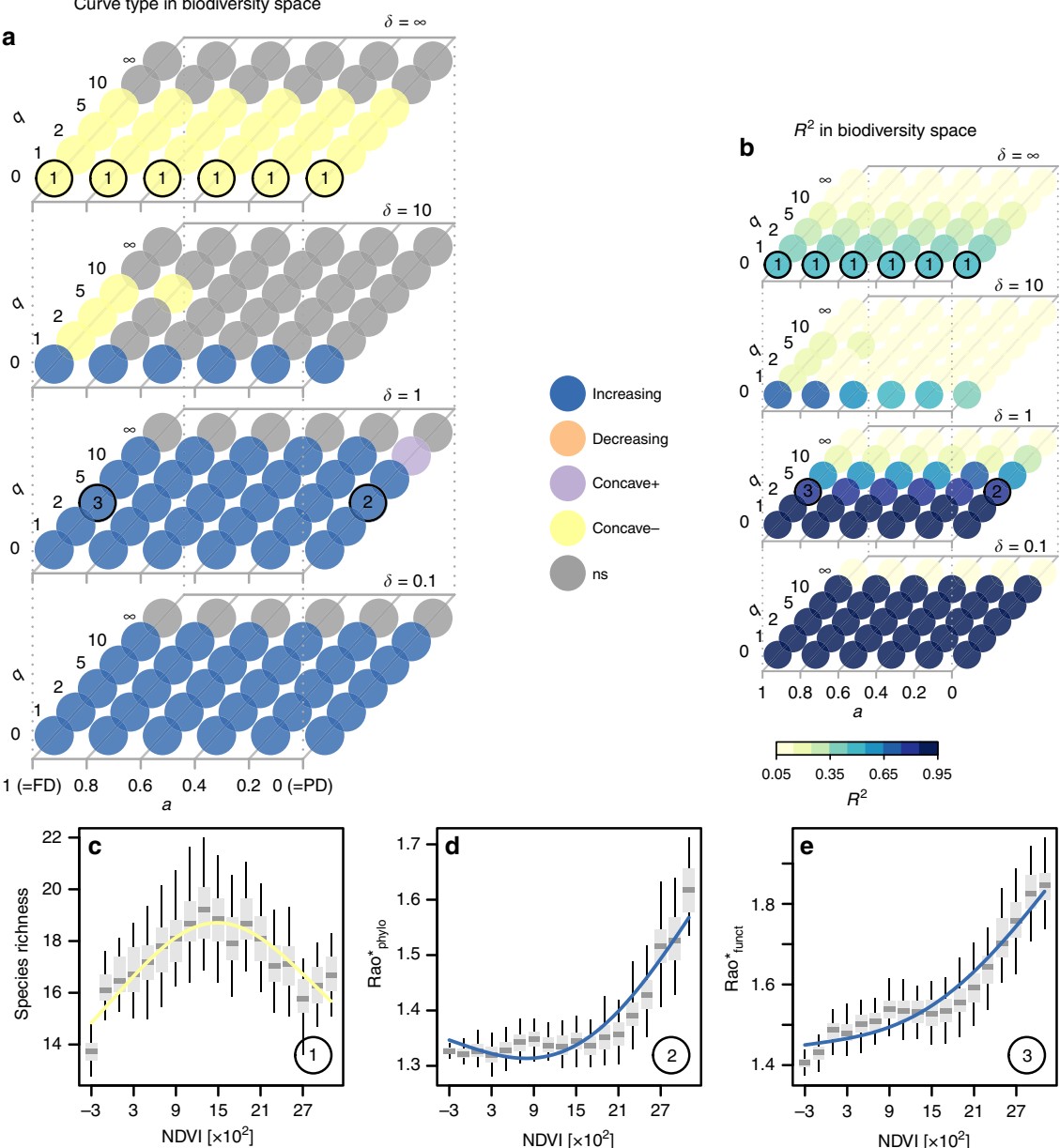

**Fig. 3 Plant biodiversity responses to NDVI across biodiversity space.** In panel (**a**) shapes of the relationships are summarized across biodiversity space. *x*-axis corresponds to increasing emphasis on trait relative to phylogenetic information; *y*-axis represent increasing emphasis on dominant relative to rare species; *z*-axis represents increasing emphasis on high versus low species similarities. Numbers indicate locations of commonly used biodiversity metrics within the biodiversity space. Panel (**b**) shows explained variance of the fits across biodiversity space. Detailed relationships for focal biodiversity metrics are illustrated at in panels (**c**–**e**) with source data provided in the Source Data file. Central lines in boxplots illustrate medians, boxes illustrate interquartile ranges, and whiskers show 95% confidence intervals. Overlaid are univariate GAM-fits with colors representing type of the curve: blue is increasing; yellow is concave-; purple is concave+; and gray is not significant. Rao* represents $1/(1-\text{Rao})$, with Rao being Rao's quadratic entropy[68].

## Results

**General relationships.** When pooling communities from all ecosystem types, we found a unimodal relationship between species richness and productivity, while most dimensions of trait and phylogenetic diversity increased with productivity (Fig. 3). We split plant community observations into 18 bins of increasing NDVI (productivity) and sampled 40 communities within each bin 100 times to calculate mean diversity for $6 \times 4 \times 6$ combinations of type of similarity ($a$), scale of similarity ($\delta$), and species dominance weight ($q$) covering the biodiversity space introduced above (Fig. 1). Using semi-parametric regression[29] and shape criteria (Supplementary Table 1), we then assessed type and explained variance of the relationships between biodiversity

measures and NDVI (see Methods). In addition to species richness, we found unimodal relationships for two thirds of the parameterizations of taxonomic diversity ($\delta = \infty$), as well as for 50% of measures considering fine scales of trait similarity ($\delta = 10$, $a = 1$). Explained variance of the model fits was highest for biodiversity measures considering coarse scales of similarity ($\delta \leq 1$) where increasing relationships prevailed (Fig. 3a, b). For biodiversity measures with increasing weights of species dominance, explained variance decreased, and for the highest dominance weight ($q = \infty$), corresponding relationships were classified as non-significant (ns).

Productivity–biodiversity relationships across all ecosystem types were largely robust when additional traits were considered

or when bioclimatic zones were compared. Relationships between NDVI and biodiversity dimensions showed little change when trait diversity was estimated based on an extended set of five traits considering also leaf dry matter content and leaf nitrogen content for a subset of the community observations (Supplementary Fig. 1). Similarly, NDVI relationships with the three focal metrics remained comparable across three bioclimatic zones (Supplementary Fig. 2). Bioclimatic zones did constrain the available productivity ranges, with colder climates being associated with lower NDVI, but where NDVI ranges overlapped, relationships were mostly equivalent. However, species richness was an exception to this rule, showing a positive association with NDVI in the colder, upper montane and lower alpine zones but a negative association in the warmer, lower montane zone.

**Effect of ecosystem type**. Within forests, productivity–biodiversity relationships were similar to those across all ecosystem types while within grasslands every significant productivity–biodiversity relationship was unimodal (Fig. 4, Supplementary Figs. 3 and 4). Forests were generally associated with higher NDVI than grasslands, with lower species richness, and with higher functional and phylogenetic diversity (Fig. 4). NDVI–biodiversity relationships within forests were unimodal for taxonomic diversity ($\delta = \infty$) while among diversity metrics focusing on intermediate and coarse scales of similarity ($\delta \leq 1$) most significant relationships were increasing (Supplementary Fig. 3). However, the goodness of fit of the relationships within forests was generally lower than for relationships across all ecosystem types. For trait diversity ($\delta \leq \infty$, $a = 1$), relationships were either non-significant or unimodal. This changed when traits were considered in isolation: the diversities of height at maturity and SM did increase with NDVI (Fig. 4c). Within grasslands, the patterns were strikingly different: unimodal NDVI–biodiversity relationships prevailed across all biodiversity dimensions (Fig. 4, Supplementary Fig. 4). Grassland biodiversity measures for which explained variance was highest included species richness and measures considering intermediate to coarse scales of trait and trait-dominated functional-phylogenetic similarity ($\delta \leq 1$, $a > 0.5$).

Within forests, productivity–biodiversity relationships appeared to be driven by the woody part of the community (Supplementary Fig. 5). Within the forest community plots investigated, slightly more herbaceous than woody species were observed, and

herbaceous species showed considerably higher Rao*$_{phylo}$ ($\delta = 1$, $a = 0$, $q = 2$), while woody species had a distinctly higher Rao*$_{funct}$ ($\delta = 1$, $a = 1$, $q = 2$). Relationships between NDVI and the focal diversity metrics of wooded forest species were equivalent to those for all forest species: unimodal for species richness, and increasing for Rao*$_{funct}$ and Rao*$_{phylo}$. The herbaceous part of the forest community, on the other hand, became less species rich with increasing NDVI and reached a minimum Rao*$_{funct}$ and Rao*$_{phylo}$ at intermediate levels of NDVI.

Productive forests had fewer species than the null expectation but phylogenetic and functional-phylogenetic diversity was mostly disproportionally high; for productive grasslands the opposite was true (Fig. 5). For forests and grasslands, we compared the diversity at the five highest NDVI bins to the diversity of random assemblages by calculating the mass fraction of the observed diversity distribution that lies above or below the null diversity distribution (density bias, see Methods). Productive forests were less diverse in terms of taxonomic diversity and for biodiversity measures considering fine scales similarity (Pagel's $\delta \geq 10$) but more diverse in measures accounting for intermediate to coarse scales of similarity (Pagel's $\delta \leq 1$). Conversely, productive grasslands showed higher species richness and lower phylogenetic and trait diversity (Pagel's $\delta \leq 10$) than null expectation.

**Effect of land use intensity**. On average the focal biodiversity metrics increased with land use intensity in grasslands, and they declined more steeply with productivity when land use intensity was high (Fig. 6). We used the average deviation of NDVI values from the seasonal signal during the growing season as a proxy for the frequency of land use practices (mowing, grazing, manuring) and split grasslands by tertiles into three levels of land use intensity. Low land use intensity was predominant for communities at the three lowest levels of productivity but negligible at the two highest levels of productivity. For all focal metrics, average biodiversity increased with land use intensity. Furthermore, NDVI-species richness relationships changed from increasing to unimodal with land use intensity, while Rao*$_{phylo}$ and Rao*$_{funct}$ shifted from unimodal to decreasing. Despite these differences in relationship types, land use intensities did not greatly affect density biases of productive grasslands throughout the whole biodiversity space (Supplementary Fig. 6).

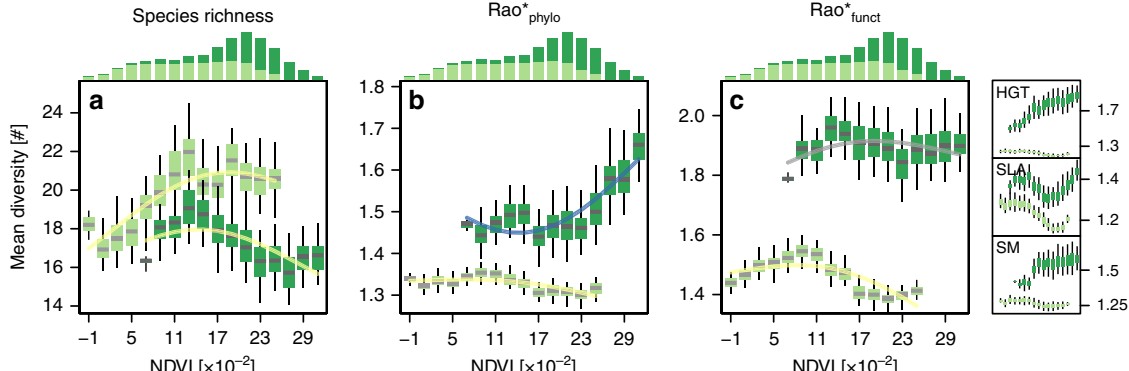

**Fig. 4 Relationships between focal biodiversity metrics and NDVI for forests and grasslands.** Productivity-species richness relationships are shown in panel (**a**); productivity-Rao*$_{phylo}$ relationships are shown in panel (**b**); productivity-Rao*$_{funct}$ relationships are shown in panel (**c**). Central lines in boxplots illustrate medians, boxes illustrate interquartile ranges, and whiskers are 95% confidence intervals for forests (dark green) and grasslands (light green). Overlaid curves are corresponding GAM-fits, colored according to curve type classification: yellow is unimodal; blue is increasing; and gray is not significant. Standard errors are hardly visible in this representation and thus not depicted. Histograms illustrate the frequency distribution of the vegetation types along the productivity gradient. Mini panels on the right show Rao* indices for traits ($a = 1$) for maximum height (HGT) diversity, specific leaf area (SLA) diversity and seed mass (SM) diversity. Rao* represents $1/(1-Rao)$, with Rao being Rao's quadratic entropy[68]. Source data for main panels are provided in the Source Data file.

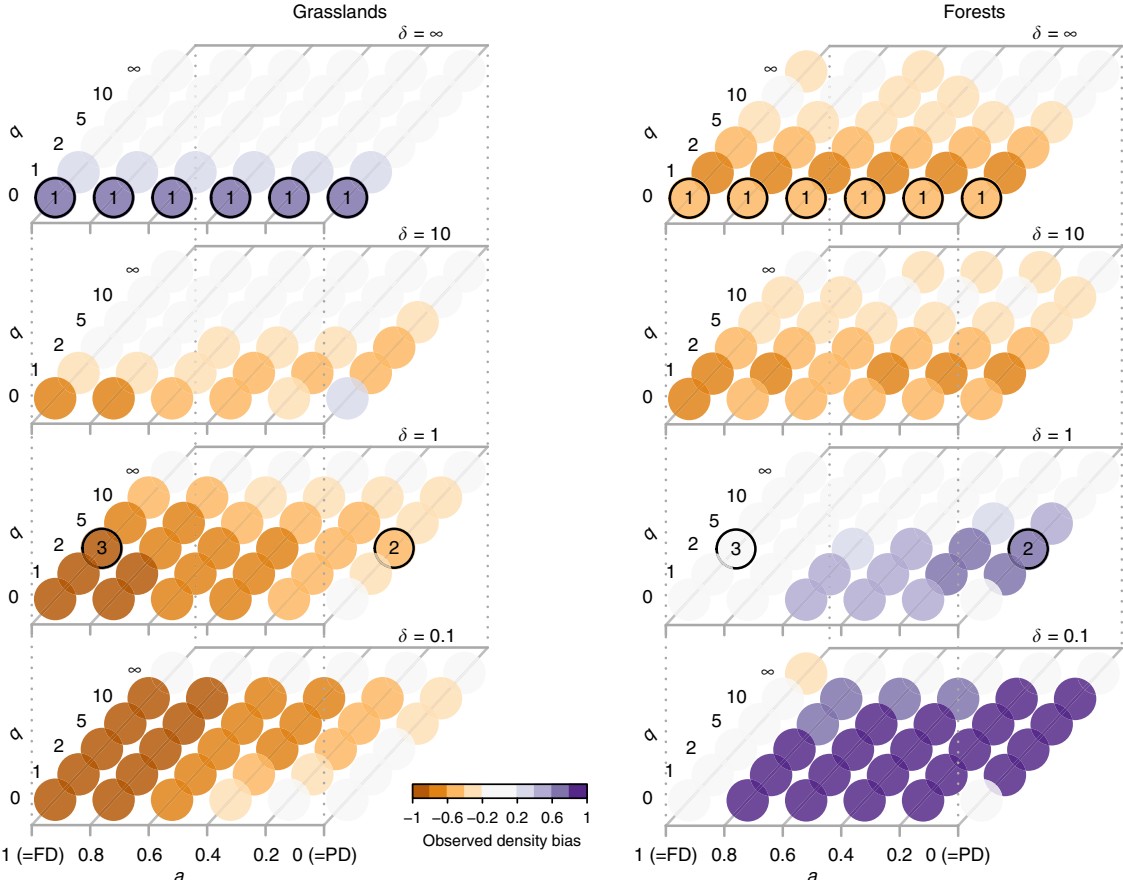

**Fig. 5 Density bias of high-productivity biodiversity.** Positive values are associated with higher biodiversity than expected from null models, negative values represent lower diversity than expected (see Methods). Results are shown for forests (**a**) and grasslands (**b**). x-axis corresponds to increasing emphasis on trait relative to phylogenetic information; y-axis represents increasing emphasis on dominant species; z-axis represents increasing emphasis on high versus low species similarities.

## Discussion

We comprehensively investigated plant productivity–biodiversity relationships by systematically evaluating different biodiversity metrics and the ecological hypotheses they imply. By varying dominance weights, and type and scale of similarity underlying biodiversity estimates, as well as bioclimatic zones, ecosystem types, and land use intensity levels, we unraveled a multitude of patterns. Our first hypothesis was confirmed: we consistently found unimodal relationships between taxonomic diversity and productivity in all ecosystem types, unless the productivity range was constrained to lower levels by cold climate or low land use intensity or to higher levels by warm climate. Consistent with the majority of below-continental-scale studies on productivity–species richness relationships in plants (summarized in ref. [3]), our results provide evidence that plant species richness generally declines at the highest productivity levels, suggesting that competitive exclusion is an important mechanism in productive plant communities[19–21,30–32]. In contrast, productivity–trait diversity and productivity–phylogenetic diversity relationships varied between ecosystem types.

Despite declining species numbers, trait and phylogenetic diversity increased in productive forests to values above random expectation. This result corresponds to the patterns expected of by the limiting similarity hypothesis[23–25]. Productivity–biodiversity relationships in forests appear to be dominated by the woody part of the vegetation (Supplementary Fig. 5), with a comparably high average longevity and thus a slow turnover[33,34]. Woody vegetation therefore forms a comparably stable, vertically structured environment, with light limitation in the lower layers, except early in

the growing season if deciduous trees form the canopy. Such environments contain limited open space for establishment but their complexity and stability allow for a high niche dimensionality, i.e., a wide variety of Grinnellian niches[22] that can be filled by specialized species. On evolutionary time scales divergent selection may then act on various traits of the present community to occupy these niche spaces, in particular if competition is high[35].

Trait and phylogenetic diversity consistently responded negatively to high productivity in grasslands, while at low to intermediate levels of productivity the relationship depended on land use intensity. Trait and phylogenetic diversity of productive grasslands was below random expectation, which was opposite to our hypothesis and to the patterns found in forests. The less stable structures in grasslands may offer less room for specialized niches, and thus trait diversity may be driven more by the interplay of productivity and disturbance filters: members of grassland communities have been shown to have similar leaf nutrient concentrations and growth rates that change with soil fertility. Soil fertility, which is linked to productivity, may thus constrain trait biodiversity in grasslands[12]. Disturbances including mowing and grazing, on the other hand, may reduce competition[36,37] and increase opportunities for establishment, leading to a diversification of traits like size, or reproductive strategy[12]. Our results indicate that disturbance induced by land use may indeed increase biodiversity, but only at low to intermediate levels of productivity. When productivity is increasing competition above a certain threshold, the disturbance filter appears to lose its effect. The identified differences between land

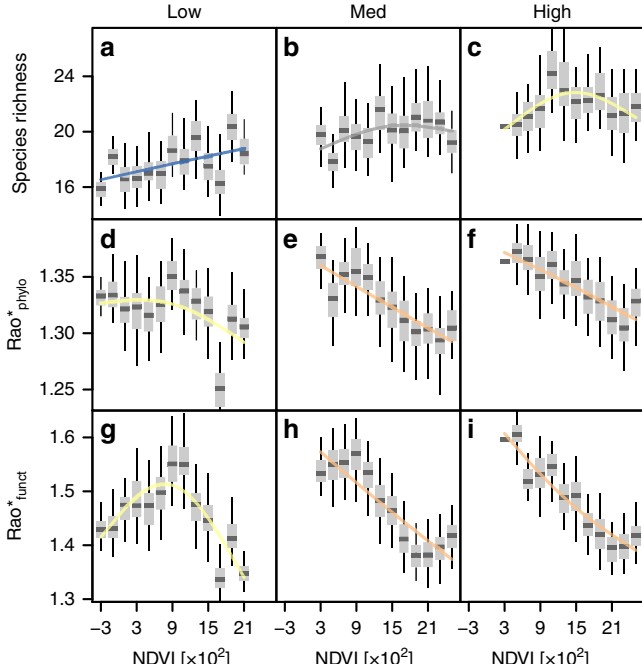

**Fig. 6 NDVI-biodiversity relationships at three levels of land use intensity in grasslands.** Relationships between productivity and species richness (**a–c**), Rao*$_{phylo}$ (**d–f**), and Rao*$_{funct}$ (**g–i**) are compared between communities of different land use intensity classes: low (**a**, **d**, **g**), medium (**b**, **e**, **h**), and high (**c**, **f**, **i**). Central lines in boxplots illustrate medians, boxes illustrate interquartile ranges, and whiskers are 95% confidence intervals. Overlaid curves are corresponding GAM-fits, colored according to curve type classification: yellow is unimodal; blue is increasing; gray is not significant; and orange is decreasing. Rao* represents 1/(1−Rao), with Rao being Rao's quadratic entropy[68]. Source data are provided in the Source Data file.

use intensities may be indicative for differences between natural grasslands, like those existing in North America or China, and more managed, semi-natural grasslands like those in Europe. However, plant trait–environment relationships can vary across continents[38] which may also affect productivity–biodiversity relationships.

The processes discussed so far not only offer plausible explanations for individual biodiversity dimensions and ecosystem types, they also fit with the overarching framework of modern coexistence theory. Modern coexistence theory[39,40] focuses on competitive communities typical for productive environments and assumes that species mainly differ along two orthogonal axes: competitive ability and niches. If co-occurring species differ more in their competitive ability than in their niches, competition will exclude inferior competitors; if they differ more in their niches than in their competitive ability, coexistence is possible. Part of the community will therefore be excluded when competition is high, leading to declining species numbers in productive environments, but not all species are affected to the same extent. The likelihood of exclusion of a species, i.e., its relative niche and competitive ability differences to other species in the community, may be mirrored in trait and phylogenetic differences[39]. Within forests trait and phylogenetic differences may mainly encode differences in the wide variety of existing niches, e.g., whether a plant is a canopy forming species or belongs to the herbaceous understory vegetation. Consequently, competition in these ecosystem types will favor species with large trait and phylogenetic differences, leading to increased trait and phylogenetic diversity.

In productive grasslands, on the other hand, species have similar growth rates, shape, and leaf chemistry. Trait/phylogenetic differences between species in these communities are therefore comparably subtle and may be more closely linked to competitive ability. So, selection for the strongest competitors in productive grasslands may lead to decreasing trait and phylogenetic diversity.

By considering plant communities from different ecosystem types and bioclimatic zones, and at various levels of land use intensity we were able to investigate productivity–biodiversity relationships at a high level of generality, but this also required making a number of limiting assumptions. Firstly, our analysis was restricted to vascular plant species and ignored the contribution of mosses or pteridophytes to biodiversity. Yet, for example in some mountainous forests, ferns and mosses may significantly contribute to the herbaceous part of the vegetation. Secondly, missing trait information constrained the analysis to about a third of all vascular plant species occurring in the French Alps. We therefore could only consider community observations with >80% coverage[41] from these most common species. Thirdly, for the over 1200 species considered, trait diversity was estimated based on three traits only. These traits are linked to key ecological functions (see Methods) and the major patterns appeared to be robust to the number of traits considered (Supplementary Fig. 1). Yet, the trait diversity used here may still not mirror all essential trade-offs acting along the productivity gradients in the various systems investigated. Fourthly, our estimates of productivity and land use intensity are proxies from remote-sensing products. The link between NDVI and productivity is well established[28,42]; the >18 years of satellite data at 30 m resolution provide the best resource available; and the strong patterns identified highlight the large amount of signal in these proxies; but estimates may still be of limited accuracy for dense forests, for communities on sites with ample bare ground, or for comparisons between contrasting ecosystem types. Finally, our approach permits the simultaneous assessment of predictions for several biodiversity dimensions, constraining the realm of plausible mechanistic explanations, but it is still empirical, and thus we cannot infer causality.

Disentangling the mechanisms underlying productivity–biodiversity relationships from binary species richness–productivity gradients alone has proven difficult. We have shown that investigating biodiversity comprehensively while accounting for bioclimatic zones, ecosystem types, and land use intensity, provides a much more detailed view on the issue, and a better test-bed for ecological theory. In the French Alps, modern coexistence theory offers a consistent explanation for the observed diversity of productivity–biodiversity relationships. Future studies investigating other systems and evaluating alternative theoretical models will be necessary to confirm the generality of this finding and pave the way to the resolution of a lasting debate in ecology.

## Methods

**Study design.** Our methodological framework consisted of a workflow (Supplementary Fig. 7) of which most steps were repeated for plant communities from all ecosystem types, forest communities, and grassland communities. In this workflow we calculated species-level functional and phylogenetic trees, adjusted their focal scale with Pagel's δ transformation[18], and converted them into distance matrices. These functional and phylogenetic distance matrices were then combined to functional-phylogenetic distance matrices using the *a* parameter[16], and provided the first input to Hill's numbers framework suggested by Leinster and Cobbold[13] with which we calculated biodiversity. The second input to this framework were either observed communities or random assemblages. We also estimated mean annual NDVI to investigate shapes and strengths of productivity–biodiversity relationships for each observed community, as well as the mean absolute deviation of NDVI from the seasonal signal during the growing season to assess the impact of land use intensity in grasslands. Finally, we compared the diversity of high-productivity environments to the diversity of random assemblages.

**Community data**. Plant community observations were provided by the French National Alpine Botanical Conservatory (CBNA) and covered the French Alps (c. 41,500 km$^2$, Fig. 2a). The initial dataset was published in ref. [43], consisted of about 43,000 observations and 3400 species of vascular plants with abundance information resolved in six coverage classes[44]. We translated these coverage classes into percentages using the conversion suggested by Münkemüller et al.[45] (Supplementary Table 2). Summed coverage estimates were allowed to exceed 100% since vegetation structure can be vertically layered.

We filtered community observations for four quality criteria. Firstly, the entire community and a minimum of ten species had to be sampled. Secondly, geopositioning information had to be available with a standard error of no more than 10 m. Thirdly, estimated total coverage had to be higher than 30%, in order to limit the distortive effect of rocks/bare soil for corresponding NDVI estimates, and lower than 250%, to discard a few observations with unrealistically high coverage values. Finally, full trait information had to be available for at least 80% of the total coverage[41]. After filtering, 11,172 community observations and 1219 species remained for multi-trait analyses, while numbers for single-trait analyses were higher (Supplementary Table 3).

**Normalized Difference Vegetation Index data**. We used the remotely-sensed Normalized Difference Vegetation Index (NDVI) to derive proxies for productivity[46] and land use intensity. Geo-corrected NDVI data were obtained from the Landsat project and had a horizontal resolution of 30 m. We downloaded NDVI estimates for all scenes (images) of the Landsat 7 satellite covering the study area from July 1999 to October 2017 and extracted NDVI values at the locations of the community plots. The extracted values were filtered for pixel quality 'clear' (no water, snow, clouds, or cloud shadow) and NDVI scores in the valid range, resulting in an average of 136 NDVI estimates per community plot (ranging from 38 to 483).

**Climate data**. We used data on bioclimatic zones and growing season to investigate the impact of climate on productivity–biodiversity relationships, and to constrain estimates of land use intensity. All climate data originated from the Climatologies at High resolution for the Earth's Land Surface Areas (CHELSA) initiative[47,48] with an original resolution of 30 arc-sec (http://chelsa-climate.org/). Bioclimatic zones were defined following Körner et al.[27] and had an original resolution of 1 × 1 km. We downscaled these values to 25 × 25 m using geographically weighted regression. Estimates on the first and the last day of the growing season (>5 °C) were based on mechanistically downscaled temperature data at 25 × 25 m resolution.

**Functional trees and traits**. We used functional trees rather than distance matrices to describe species trait similarity in order to be able to adjust the scale of similarity with Pagel's δ parameter[18]. Functional trees were built on trait information including the key traits SLA, height at maturity (HGT), and SM. SLA, HGT, and SM are tightly linked to the fundamental life missions resource acquisition, survival, and reproduction[49], and thus useful to capture variation in plant ecological strategies:[50,51] SLA mirrors the trade-off between fast resource uptake and long lifespan; SM represents the trade-off between fecundity and energy invested per offspring individual;[52] and HGT is related to competitive ability and avoidance of environmental stress[53]. Furthermore, in a sensitivity analysis we also included the leaf-spectrum-traits leaf dry matter content and leaf nitrogen content (LDMC and LNC, respectively)[54]. These traits were available for 300 species which constituted >80% of coverage in 3300 community plots. Additionally, we used information on plant woodiness in order to distinguish the herbaceous from the woody species in forest communities. Trait data originated from Thuiller et al.[43].

Functional dendrograms were derived from the combined information of SLA, HGT, and SM, and for each trait individually. First, all traits were log-transformed and scaled to unit variance. For species with complete trait information we then calculated distance matrices using the Euclidean distance measure and finally derived hierarchical clusters with the unweighted pair group method with arithmetic mean (UPGMA) algorithm to produce ultrametric, functional dendrograms[55].

We checked to which degree the functional variation was hierarchical by applying Mantel tests, comparing phenetic distances (obtained from the dendrograms) with functional distances from the distance matrices[56]. Based on 9999 randomizations, we found significant ($p \le 0.001$) correlations of 0.74, 0.80, 0.80, and 0.73 for functional trees based on all traits, SLA, HGT, and SM, respectively. A substantial fraction of the functional variation was therefore preserved in the trees.

**Phylogenetic tree**. We used an ultrametric genus-level phylogeny of plants occurring in the European Alps and extended it with random binary trees to obtain a species-level resolution. The original tree was developed by Thuiller et al.[43] based on Genbank sequences and included 947 genera, covering 99% of the species for which trait information was available (Supplementary Table 3). In order to bring the tree to species-level resolution, we randomly generated as many binary splits as were necessary to link the genus-level tree tips to all observed species within each genus, using the Yule model as implemented in the R package apTreeshape[57]. We

repeated this procedure to create 100 and 2000 possible species-level trees and used them to calculate phylogenetic and functional-phylogenetic diversity of resampled sets of observed and null communities, respectively (see subsections Calculating biodiversity and Constructing null communities). Limited phylogenetic resolution had little effect on our analyses. Indeed, across the filtered community dataset, on average only 1% of genera were represented by three or more species in local communities and thus not fully resolved.

Phylogenetic distances were not significantly correlated with functional distances. The phenetic distances obtained from the functional dendrogram and ten replicates of species-level phylogenetic dendrograms showed non-significant Mantel correlations (all $p > 0.05$) of $0.004 \pm 0.000$, when on 999 randomizations were used (see also Supplementary Fig. 8).

**Assigning ecosystem types to community plots**. Ecosystem type was directly reported only for an insufficient fraction of the community plots. We therefore derived this information a posteriori based on the typical habitat of the majority of the plant coverage present. Two approaches may be used to add ecosystem type information in retrospect: obtaining land cover information for the coordinates of the community plots (e.g., from high-resolution remote-sensing products) or by inferring the ecosystem type from the species composition of the community plot. We used the latter, more direct approach by deriving for each species whether it typically or occasionally occurred in forests or grasslands, or both ecosystem types, using information from Flora Indicativa[58]. We then defined coverage criteria to determine whether observations represent forest communities ($n = 3324$), grassland communities ($n = 3738$) or none of the above ($n = 4110$) (Supplementary Tables 3 and 4).

**Estimating productivity and land use intensity**. We approximated productivity (for each community observation) and land use intensity (for grassland observations) from irregular NDVI time-series as observed by the Landsat program. To this end, we used generalized additive models (GAMs)[29]. The theoretical range of NDVI values is bounded between −1 and 1, but the observed values only covered the central part of that range (−0.04 to 0.32). Our data were therefore hardly affected by these boundaries and it was feasible to assume that NDVI values follow a Gaussian error distribution. To approximate productivity, we fitted NDVI as a function of Julian day, using a cyclic cubic regression spline with a flexibility of five degrees of freedom at maximum ($k = 5$), and year of measurement, using a factor term. From these fits, we then predicted mean interannual NDVI. For land use intensity, we fitted similar models, but only considering the observations taken during the growing season, where we assumed the majority of land use practices to take place. Within this period, we fitted NDVI as a function of Julian day using ordinary thin plate regression with a flexibility of three degrees of freedom at maximum ($k = 3$). We assumed that in semi-natural grasslands the model error remaining after correcting for the seasonal signal is mainly driven by land-use practices including mowing, grazing and fertilizing, and used the mean absolute residual of the model fits as a proxy for land use intensity[59]. All GAMs were fitted using the R package mgcv[60].

**Binning plant community observations**. Before estimating biodiversity, we split plant community observations into bins defined by distinct combinations of NDVI, ecosystem type, bioclimatic zone, and land use intensity. Investigating biodiversity within discrete NDVI bins allows determining uncertainty via bootstrapping. The NDVI gradient in the data was split into 18 bins (−0.04 − −0.02, −0.02 –0.,…, 0.3 –0.32) in order to enable comparison of several productivity levels while still having sufficient observations per bin. For ecosystem type, we either pooled all ecosystem types, or we distinguished forest and grassland observations. Furthermore, we distinguished the lower montane, upper montane, and lower alpine bioclimatic zones, as well as three levels of land use intensity in grasslands (separated by tertiles of mean absolute deviations from the seasonal NDVI trend). From the community observations within each bin, biodiversity was estimated for 100 bootstrap samples of 40 observations. When land use intensity levels were compared in grasslands, bin-wise biodiversity was estimated for 100 bootstrap samples of 20 observations, due to the reduced number of observations per bin. Observations were sampled randomly under the constraint that they had to be at least 5 km apart from each other. Binning with regular bin sizes leads to smaller sample sizes at the boundaries of the productivity range, which may cause undesired edge effects. We tested for such effects by using an alternative binning criterion of 5%-percentiles of the productivity data and found edge effects to have a small impact on our results (Supplementary Fig. 9).

**Calculating biodiversity**. We calculated biodiversity by successively defining scale of similarity, type of similarity, and dominance effect. To adjust the scale of similarity, we transformed functional and phylogenetic trees with Pagel's δ transformation[18]. This transformation was originally developed in a phylogenetic context, but can also be applied to functional dendrograms[15]. Transformations with δ below 1 increasingly inflate deep, close-to-root branches of the dendrogram, emphasizing differences between distinct groups, while transformations with δ above 1 increasingly inflate the shallow, close-to-tip, branches, focusing on fine differences like those between sister species. If δ approaches infinity, the

dendrogram becomes rake-like, assuming equal similarity between all species, and corresponding biodiversity metrics approach taxonomic diversity (see also Fig. 1). We estimated biodiversity for $\delta$ values of 0.1, 1, 10 and $\infty$.

Once the scale of similarity was set, type of similarity was determined based on the functional-phylogenetic weighting parameter[16] $a$. To this end, $\delta$-transformed functional and phylogenetic trees were converted into distance matrices, and used to calculate a combined matrix of Euclidean functional-phylogenetic distances based on the formula

$$\text{FPDist} = \left( a\,\text{PDist}^2 + (1-a)\text{FDist}^2 \right)^{\frac{1}{2}}, \qquad (1)$$

where PDist represents the phylogenetic distance matrix and FDist the functional distance matrix. Functional-phylogenetic distances can be considered functional distances that account for information from unmeasured, phylogenetically correlated traits, and may contain information not apparent from phylogenetic or functional distances alone[16]. Biodiversity was calculated for $a$ values of 0, 0.2, 0.4, 0.6, 0.8 and 1.

Finally, the importance of species dominance was defined through the parameter $q$ in Hill's numbers frame work[14]. To this end, the obtained functional-phylogenetic distance matrices were converted into a similarity matrices (one minus scaled distance) and used as similarity information in the framework of Leinster and Cobbold (2012)[13] to calculate biodiversity. Besides considering similarity information, in this framework the importance of dominance can be varied by setting the dominance weight $q$, which is bounded by zero and infinity[13]. We calculated biodiversity for Hill's numbers of 0, 1, 2, 5, 10, and $\infty$, using the R function abgDecompQ provided by Chalmandrier et al.[15].

**Modeling productivity–biodiversity relationships**. We investigated productivity–biodiversity relationships using GAMs. Our biodiversity estimates are effective numbers, i.e., non-negative but not necessarily integer values which should not be modeled assuming Gaussian error distributions. However, Poisson error distribution, the typical alternative for count data, was also not feasible as it requires integer values. We therefore log-transformed biodiversity estimates prior to model fitting with the Gaussian error assumption. We fitted the response to productivity as a smooth term with a restricted flexibility of three degrees of freedom at the most ($k = 3$).

**Classifying productivity–biodiversity relationships**. We generated response curves from the productivity–biodiversity model fits and used goodness of fit and curve shape criteria to separate five classes of curves. Goodness of fit was determined based on $R^2$, which here represents the explained variance when fitted biodiversity estimates were back-transformed to the original scale. Model fits with $R^2 < 0.15$ were assumed to be insignificant and corresponding curve types labeled 'ns'. We used this relatively strict threshold, in order to exclude noisy relationships from interpretation. Response curves were predicted within the range of NDVI observations. Curves were considered concave− (unimodal) if they had their minima at the minimum or maximum of the prediction range and their maxima somewhere within the prediction range. Correspondingly, to fall into the class concave+, curves had to have their maxima at the minimum or maximum of the prediction range and their minima somewhere within the prediction range. However, our relatively inflexible GAM fits by tendency fell within this strict definition of concave + for biodiversity estimates that increased exponentially with productivity. Due to this artifact and since there was no theoretical reason to expect concave+ productivity–biodiversity relationships, we added an additional criterion for this class: curves had to significantly increase on both sides of the minimum. This was the case if biodiversity predictions at both edges of the prediction range surpassed 25% of the predicted biodiversity range. If this was not the case, and maximum biodiversity was predicted for highest NDVI, curves were assigned 'increasing'; If this was not the case and maximum biodiversity was predicted at lowest NDVI, curves were assigned 'decreasing'. Curve shape criteria are summarized in Supplementary Table 1.

**Constructing null communities**. We constructed null communities to investigate how biodiversity in high-productivity environments differs from random expectation. Null communities were constructed for all ecosystem types, forests only, grasslands only, and land use intensity levels within grasslands based on an extension of Gotelli's swap algorithm[61,62] for abundance data. In a first step, we applied this former algorithm to the presence/absence version of our species by site matrix: random pairs of present species and sites were repeatedly chosen and species presence was swapped. While constructing random communities, this approach preserves local species richness per site and among site species occurrence frequencies. In a second step, we randomly assigned the observed dominance values at each site to the new species lists. This way, also local abundance distributions per site were preserved. For biodiversity analyses 2000 times 40 of these null communities were drawn (2000 times 20 for null models specific to land use intensity levels) under the constraint that sites from each productivity bin had to be represented in equal fractions. For the derived null communities we then calculated all dimensions of biodiversity as described in subsection Calculating biodiversity.

**Comparing observed biodiversity with null biodiversity**. For forests and grasslands, we compared biodiversity dimensions in the highest productivity bins with the corresponding biodiversity dimensions of null communities. We approximated the frequency distributions of null and observed biodiversity by crude, continuous density functions (adjust = 2 in the density command in R[63]) and used them to calculate the mass fraction of the observed biodiversity density distribution that lies at higher or lower biodiversity than the null biodiversity density distribution. If the surplus of the observed density distribution was at higher biodiversity, the resulting fraction was multiplied with +1, otherwise it was multiplied by −1. We calculated these biases individually for the five highest productivity bins (the three highest bins for estimates specific to land use intensity levels in grasslands) and report their averages.

**Reporting summary**. Further information on research design is available in the Nature Research Reporting Summary linked to this article.

## Data availability

NDVI data from the Landsat mission. Climate data from CHELSA. Plant community data, phylogenetic data, and trait data are available from Wilfried Thuiller (wilfried.thuiller@univ-grenoble-alpes.fr) on reasonable request. The source data underlying the main results (Figs. 2a, 3c–e, 4, and 6) are provided as a Source Data file.

## Code availability

All analyses were conducted in the R environment[63] (version 3.4.2). The packages used include ade4[64] (version 1.7-13), ape[65] (version 5.2), apTreeshape[57] (version 1.4-5), geiger[66] (version 2.0.6), mgcv[60] (version 1.8-20), and raster[67] (version 2.8.19). Code generated for analyses and illustrations is available from the corresponding author.

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

## Acknowledgements

This work was supported by the ANR-SNF bilateral project OriginAlps, with grant numbers 310030L_170059 (P.B., N.E.Z) and ANR-16-CE93-004 (W.T., S.L., T.M.). The authors further acknowledge Swiss National Science Foundation (SNF) grant numbers 1003A_149508 (N.E.Z.), and 31003A_173342 (C.H.G.). W.T. and T.M. acknowledge support from the 'Investissement d'Avenir' grants managed by the ANR (Trajectories: ANR-15-IDEX-02; Montane: OSUG@2020: ANR-10-LAB-56). Moreover, the authors thank Isabelle Boulangeat for providing the stickTips function to calculate random binary trees; Dirk Karger, Julien Renaud, and Maya Gueguen for preparing the climate data; and Achilleas Psomas for his support in estimating land use intensity. We also thank the Conservatoire Botanique National Alpin for its continous effort in collecting high quality vegetation plots.

## Author contributions

P.B., W.T. and N.E.Z. conceived the general idea, and designed the study with the help of C.H.G, T.M., L.P. and S.L. P.B. performed the analysis and led the writing of the manuscript. All authors significantly interpreted results and contributed to writing and editing.

## Competing interests

The authors declare no competing interests.
