## [Peer Review File · Nature Communications]

Reviewers' Comments:

Reviewer #1:

Remarks to the Author:

This is a very nice paper of broad relevance for the ecosystem functioning-biodiversity research field. There are several novel aspects, in particular the introduction and then use of different biodiversity measures, showing that at high-productivity sites functional/phylogenetic diversity can be increased due to greater biotope space, allowing more species niches to fit in.

The paper uses a large data set of sample surveys and analyzes them with state-of-the-art statistical methods. It is very well written, so I only have some general points:

1) I would strongly recommend to change "biodiversity-productivity" to "productivity-biodiversity" in the title and throughout the paper. Although in sample surveys we deal with correlations rather than causation, the fitting of a unimodal relationship is only possible if productivity is the x and biodiversity the y, therefore I would use "x-y relationship". The very very large literature about biodiversity-ecosystem functioning or biodiversity-productivity relationships (BEF studies) is considering biodiversity as x and productivity as y. These different possibilities also relate to the explanatory theories mentioned on line 270. The first group tries to explain the causal relationship productivitybiodiversity and the second group tries to explain causality in the opposite direction (and of course both directions will occur in nature, but in an experiment the first must manipulate e.g. site fertility and the second e.g. functional richness). This point is pervasive throughout the MS and should be considered throughout in a revision.

2) Most data probably come from more or less managed ecosystems. This is mentioned, but the possible consequences are not deeply discussed. Compared to other areas such as more natural grasslands in America or China or more natural forests in some subtropical and tropical regions different results might be expected.

3) There is one more dimension of biodiversity that may be implicit in the paper but I could not find out: often, when papers report relationships from forests, they only consider the biodiversity of trees (e.g. > 5 or 10 cm DBH). Was this also the case here? If not, should it be added? And how would mosses and other cryptogams change grassland relationships (or were they included as plant species?)? I'm not saying this must be included, but at least I would point out that this would be an additional dimension (to use subsets of taxonomic groups to calculate the different diversity measures). Furthermore, it would make understanding easier if dominance would be linked to the evenness dimension of biodiversity, which is the more familiar and "positive" term in the context. Maybe it would also be good to justify the use of Rao's Q, which has the advantage and disadvantage that it is not correlated with species richness (whereas other FD measures such as that of Petchey & Gaston have the advantage/disadvantage that they are correlated with richness).

4) I understand that there are advantages of not correcting for any covariates, but it does require a bit of justification, because generally a problem of such data is that x-y relationships could be caused by third variables affecting both and thus leading to an apparent relationship.

5) The comparison between grasslands and forests is not well developed and missing in the Introductory Paragraph where it could replace the final sentence (which is not really new, even though - contrary to common "feeling" -- it has received little concrete empirical support so far during the past 25 years of BEF experiments). The authors do say that they don't have a concrete expectation, but they report clear differences, which asks for more insightful discussion, perhaps related to point 3) above.

6) Technically, would it be possible to get a better resolved phylogeny that would at least go to species level on some branches? This would be interesting because the z-dimension of biodiversity is related to it.

Some minor points:

21: "niche dimensionality" or "biotope space"?

23-24: "likely reflecting ..." is not elaborated later on (and I think there must be other, better explanations).

40-42: Allan et al., Ecology 2013 may be interesting in this context.

151: I would write "low versus high species similarities" to be consistent with Fig. 1.

Reviewer #2:

Remarks to the Author:

Rev. Brun et al. The biodiversity –productivity relationship varies across diversity dimensions. Nature Communications

This is a very interesting piece of work in which one of the most relevant and long standing questions in Ecology is evaluated, the diversity to productivity relationships in plant communities. Their approach is built on two basic pillars, the explicit consideration of two complementary diversity dimensions, the phylogenetic and functional ones historically forgotten, and a new supposedly comprehensive approach based on the consideration of the dominance, type of similarity between species and the scale of this similarity. I like especially the effort of building theory by using observational but massive data sets, a type of information usually very badly treated in outstanding scientific publications.

Although I do think the paper is valuable, there are some questions that, from my perspective, are very hard to follow and understand. For instance, the simultaneous use of different types of diversities measured directly on the field is clear that could inform of different mechanisms of assembly which could give new insights to this relationship with productivity; however here, authors have systematically transformed their original diversity values to evaluate the effect of the variation of these three elements, type and scale of similarity and dominance. Why? This is a sort of sensitivity analysis, but not explicit. This is the heart of the paper and, unfortunately, it is not properly justified and explained. For instance, instead of transforming your data set by using a complete set of Hill's numbers you could group your original information as a function of the realized dominance and, lately, evaluate the primary productivity. As I comment above your approach seems a mechanism for evaluating the sensitivity of these numbers. Even more, I think that most readers would be expecting a framework in which the three metrics of diversity, taxonomic, phylogenetic and functional were compared. This is not exactly the case, so, I think authors should better explain why they prefer their very novel approach.

Although the workflow is really impacting and well developed I feel that the theoretical framework is weaker. For instance, do you really think that the information provided by the functional diversity facet and the phylogenetic one is directly connected? Or, that the prevalence between one or the other is only a question of spatial scale? If so, what you were saying is that the really relevant information is the phylogenetic although at small scales, the functional one, could be a very efficient alternative. This needs a clearer information.

On the other hand, I miss also some more information about the relevance of dominance to understand these relationships. Why dominance? I can imagine that other surrogates of primary

productivity could be also very relevant, for instance the total cover.

I have also many difficulties with the reduced number of functional traits you have used. Do you think this number is high enough? Even more, do you think they are critical enough to explain the diversity of vegetation types you found. For instance, SLA and height seems only relevant in light limited communities. In this sense, seed mass is critical for dispersal and establishment, but, do you think there is some kind of limitation for seed dispersal and early establishment in your region of reference? Even more, you are using central values, but don't you think other terms related to the distribution of these traits in the population could be also of interest? If so, you could include not only the mean value, but, for instance the Standard Deviation per species.

Obviously this is more work, but, at least, you should try to better explain these questions. In this sense I miss also to know if the three considered traits have a phylogenetic signal or not.

I have also some problems with the field sampling work. For instance, what is the size and the shape of the vegetation samples? Since the number of species in your graphs is very small, I guess they are very small at least for forest or shrubby plan communities. How was the cover per species estimated? Where were the plots located? Was cover per species estimated by a unique technician?

In your vegetation data set, a very different type of plant communities is included. The most shocking differences are related to the fact that you included natural communities with a very low-level of disturbance together with plant communities submitted to regular and intense man-driven management such as cutting or manuring in some semi-natural grasslands. Obviously this implies drastic and dramatic differences, especially in the case of some grasslands. From my perspective this can obscure your results and conclusions. In addition, some of your vegetation groups are reasonably homogeneous, such as forests, but other are very internally heterogeneous. For instance, within grasslands you can find intensively managed vegetation types together with almost pristine and untouched pastures like those of some alpine zones. In my opinion this makes inferring conclusions very complex. As a recommendation I suggest to organize your data set with some estimates of man-driven perturbation levels.

Although of less importance I also miss some more information on the theoretical background of the so-called Theory of Coexistence. What are you talking about? You only briefly introduced the idea of filtering and limiting similarity, but as you know patterns of functional clustering or overdispersion can be obtained through very different mechanisms. If so, how could you assign an observed functional pattern and structure to a specific mechanism. This is critical if you want insert your discussion in this theoretical background.

An additional problem is related to the fact that forests in French Alps are very poor at least in the tree guild, so observed dominance in them is very intense. Comparison with other vegetation types is then, very complex with vegetation plots structured in an almost binomial structure, very dominated (forests) to lowly dominated (the rest).

We need some more information about the binning of the data set and the selection of 18 bins of increasing NDVI.

At the end of the first paragraph of the Discussion section there is a sentence that says nothing, "In the following paragraphs we compare the patterns with the recent literature.....".

I have some difficulties with the dichotomy you posed on the table in relation to the connection between niche separation and competitive ability. As you suggested, these two ideas are orthogonal but in my opinion, niche separation is closely related with your, as species, ability to compete. Probably I am wrong, but a more detailed explanation will be acknowledged.

A very minor concern is that I do think the annual plant communities should be eliminated from the final data set. The problem is that the NDVI of this plots should be restricted only to the season in

which these plants were monitored. IN this sense, minor questions, I miss some more information about the way in which numerical classifications are done. The only thing it is commented in the text is that ".....derived hierarchical clusters" There is no information of the algorithm. The rule of thumb for considering an acceptable model of $R^2 < 0.15$ is too generalist. Could you include some more information to support this point?

Reviewer #3:

Remarks to the Author:

The paper examines the relationships between species richness, trait diversity or phylogenetic diversity and productivity. Analyses of species richness vs productivity and a multi-faceted measure of trait-phylo-diversity vs productivity are conducted. The team finds that these relationships are different and species richness-productivity relationships are not enough to really understand biodiversity ecosystem function relationships. I suppose the ultimate claim her (which summarizes the first paragraph of the paper and the paper generally) is not really that novel. It reminds me of the work by Cadotte et al. (PNAS and PLoS One papers) from about a decade ago and even Tilman et al. from two decades ago (Tilman et al. 1997 Science). That is, we have realized for quite some time that there is a lot more to B-EF research than simple plots of species richness vs some variable.

The novel twist of this paper may be the 3-d mixing of traits-phylogeny-dominance. It blends Cadotte's framework using the α parameter, pagel's tree transformations in a somewhat interesting way. Thus, from a methods perspective, this makes some kind of advance on how we may compute something new. However, clarifying how this new measurement leads us to new biological insights is quite unclear.

This brings me to the crux of my critique. The approach largely explores parameters space in a large dataset. The hypotheses set forth at the end of the introduction that are linked to a particular pattern are certainly not the only hypotheses that could give the pattern described. Thus, at the end we have a statement that parameter values 1,2 and 3 fit the data the best, but we really don't know why and strong statements supporting one model (e.g. stress gradients) or another (e.g. limiting similarity) are not reasonable.

In sum, the main conclusion being drawn is not all that different from key take home messages from Tilman and Cadotte in the past and the new analytical approach does not provide new biological insights. I consider the work solid, but unlikely to move the field forward.

Detailed response to comments of Referee 1

R1.1:

This is a very nice paper of broad relevance for the ecosystem functioning-biodiversity research field. There are several novel aspects, in particular the introduction and then use of different biodiversity measures, showing that at high-productivity sites functional/phylogenetic diversity can be increased due to greater biotope space, allowing more species niches to fit in.

The paper uses a large data set of sample surveys and analyzes them with state-of-the-art statistical methods. It is very well written, so I only have some general points:

1) I would strongly recommend to change "biodiversity-productivity" to "productivity-biodiversity" in the title and throughout the paper. Although in sample surveys we deal with correlations rather than causation, the fitting of a unimodal relationship is only possible if productivity is the x and biodiversity the y, therefore I would use "x-y relationship". The very very large literature about biodiversity-ecosystem functioning or productivity-biodiversity relationships (BEF studies) is considering biodiversity as x and productivity as y. These different possibilities also relate to the explanatory theories mentioned on line 270. The first group tries to explain the causal relationship productivitybiodiversity and the second group tries to explain causality in the opposite direction (and of course both directions will occur in nature, but in an experiment the first must manipulate e.g. site fertility and the second e.g. functional richness). This point is pervasive throughout the MS and should be considered throughout in a revision.

This is a good point, thank you. We have changed "biodiversity-productivity" to "productivity-biodiversity" relationships throughout the manuscript.

R1.2:

2) Most data probably come from more or less managed ecosystems. This is mentioned, but the possible consequences are not deeply discussed. Compared to other areas such as more natural grasslands in America or China or more natural forests in some subtropical and tropical regions different results might be expected.

It is true that one potential of our data set, which we had not tapped into in the initial submission, is that we have community observations across a range of land use intensities, in particular for grasslands. In an additional analysis we have now grouped our grassland observations by three levels of land use intensity (see R2.5) and investigated productivity-biodiversity relationships individually within each level. Based on these findings we now discuss the role of land use intensity in grasslands in more depth (Lines 296-314). As a side note, we had removed all plantation forest plots prior to analyses and only kept semi-natural and natural ones.

R1.3:

3) There is one more dimension of biodiversity that may be implicit in the paper but I could not find out: often, when papers report relationships from forests, they only consider the biodiversity of trees (e.g. > 5 or 10 cm DBH). Was this also the case here? If not, should it be added? And how would mosses and other cryptogams change grassland relationships (or were they included as plant species?)? I'm not saying this must be included, but at least I would point out that this would be an additional dimension (to use subsets of taxonomic groups to calculate the different diversity measures).

Our forest biodiversity estimates are based on vascular plant species including all sizes of woody and herbaceous vegetation (now explicitly mentioned in the Data section of the Methods and in the Introduction). In Supplementary Fig. 5*, we now show productivity-biodiversity relationships for the herbaceous and the woody parts of forest communities separately, indicating that overall relationships follow the productivity-biodiversity relationships of the woody part of the vegetation (see also 225-233 in the results). Furthermore, we now added a paragraph on limitations to the discussion in which we mention (among other things) the issue of not considering mosses and cryptogams (Lines 337-358). In grasslands of low altitudes, we'd expect only a minor effect from mosses and lichens. In high altitude sites and in forests, the more abundant mosses and lichens may or may not affect the found relationships.

Furthermore, it would make understanding easier if dominance would be linked to the evenness dimension of biodiversity, which is the more familiar and "positive" term in the context. Maybe it would also be good to justify the use of Rao's Q, which has the advantage and disadvantage that it is not correlated with species richness (whereas other FD measures such as that of Petchey & Gaston have the advantage/disadvantage that they are correlated with richness).

We now link the concept of dominance to evenness in the introduction (Lines 54-57). Furthermore, we justify the use of exemplary indices linked to Rao's entropy due to their complementarity with species richness (Lines 150-154). However, we also emphasize in a second statement that these exemplary indices were merely picked to facilitate presentation and that all major results are also provided across biodiversity space, either in the results or in the Supplementary information.

R1.4:

4) I understand that there are advantages of not correcting for any covariates, but it does require a bit of justification, because generally a problem of such data is that x-y relationships could be caused by third variables affecting both and thus leading to an apparent relationship.

We did not include additional covariates as predictors for biodiversity here, but we corrected for the major additional drivers by sub-setting community observations into different classes. We had done this for ecosystem types in the initial submission. Now, we have also done it for bioclimatic zones (Supplementary Fig. 2*, Lines 192-198) and land use intensity in grasslands (Fig. 6* and supplementary Fig. 6*, Lines 252-262, see *R2.5*). In the resubmitted version of the manuscript, we therefore consider and discuss these major additional drivers, and this makes

the study more comprehensive. Yet, we acknowledge that based on this observational set-up it is not possible to infer causality with certainty. We now also mention this issue in the limitations paragraph of the discussion (Lines 337-358).

R1.5:

5) The comparison between grasslands and forests is not well developed and missing in the Introductory Paragraph where it could replace the final sentence (which is not really new, even though - contrary to common "feeling" -- it has received little concrete empirical support so far during the past 25 years of BEF experiments). The authors do say that they don't have a concrete expectation, but they report clear differences, which asks for more insightful discussion, perhaps related to point 3) above.

We now extended our discussion on the differences between forests and grasslands (Lines 284-314). In addition, we shortened the abstract to 150 words to meet the format requirements, removing the final sentence.

R1.6:

6) Technically, would it be possible to get a better resolved phylogeny that would at least go to species level on some branches? This would be interesting because the z-dimension of biodiversity is related to it.

This is a good point. We are currently developing a species-level phylogeny using molecular data for all species of the Alps. However, unfortunately we are still too far from being able to use/release it. Nevertheless, for this study the genus-level phylogeny is already quite insightful, as across all ecosystem types on average only 1% of genera were represented by three or more species in local communities and thus not fully resolved. We now also mention this fraction of observed genera with lacking resolution in the methods (Lines 460-463).

R1.7:

Some minor points:

21: "niche dimensionality" or "biotope space"?

We believe that "niche dimensionality" is the more appropriate term here, as it refers to habitat complexity and the potential number of Grinnellian niches available. It also links more nicely to coexistence theory. "Biotope space", on the other hand, appears to be linked to geographic space which is not what was meant in the statement.

We removed the term from the Abstract due to the word constraint but now elaborate on it in more detail in the discussion (Lines 291-293).

23-24: "likely reflecting ..." is not elaborated later on (and I think there must be other, better explanations).

We removed the phrase.

40-42: Allan et al., Ecology 2013 may be interesting in this context.

Thanks for the hint. We now incorporate the findings from this study into the introduction.

151: I would write "low versus high species similarities" to be consistent with Fig. 1.

We replaced this description of δ in in the legends of all figures where it applied.

Detailed response to comments of Referee 2

R2.1:

This a very interesting piece of work in which one of the most relevant and long standing question in Ecology is evaluated, the diversity to productivity relationships in plant communities. Their approach is built on two basic pillars, the explicit consideration of two complementary diversity dimensions, the phylogenetic and functional ones historically forgotten, and a new supposedly comprehensive approach based on the consideration of the dominance, type of similarity between species and the scale of this similarity. I like especially the effort of building theory by using observational but massive data sets, a type of information usually is very badly treated in outstanding scientific publications.

Although I do think the paper is valuable, there are some questions that, from my perspective, are very hard to follow and understand. For instance, the simultaneous use of different types of diversities measured directly on the field is clear that could inform of different mechanisms of assembly which could give new insights to this relationship with productivity; however here, authors have systematically transform their original diversity values to evaluate the effect of the variation of these three elements, type and scale of similarity and dominance. Why? This is a sort of sensitivity analysis, but not explicit. This is the heart of the paper and, unfortunately, it is not properly justified and explained. For instance, instead of transforming your data set by using a complete set of Hill's numbers you could group your original information as a function of the realized dominance and, lately, evaluating the primary productivity. As I comment above your approach seems a mechanism for evaluating the sensitivity of these numbers. Even more, I think that most readers would be expecting a framework in which the three metrics of diversity, taxonomic, phylogenetic and functional were compared. This is not exactly the case, so, I think authors should better explain why the prefer their very novel approach.

Taxonomic, phylogenetic and functional diversity are important dimensions that potentially inform about different mechanisms of assembly. However, it is important to notice that there are no "original biodiversity values" but only information on the species present, their coverage, their position on a phylogenetic tree and their similarity in functional traits, as well as countless equations to boil this information down into one number of biodiversity. As Referee 1 points out, there are FD measures correlated to species richness as well as FD measures that are

uncorrelated to species richness (*R1.3*) which will have different relationships with productivity. Yet, unless one pursues a very specific research question, there are very few criteria to know which measure is most appropriate. With our approach, we aimed at avoiding an arbitrary decision for one measure over another and instead studied the productivity-biodiversity relationships systematically for a broad array of biodiversity indices including most of the commonly used ones. This approach allowed us to reach conclusions that are independent of one particular biodiversity index and thus are more general.

We tend to disagree that this approach is a sort of sensitivity analysis, but it is rather a comprehensive analysis of biodiversity across a profile of dominance importance and species similarity. In other words, the different metrics are not estimated to quantify uncertainty in the shape of a universal productivity-biodiversity relationship, but rather they are associated with specific expectations and elegantly can be computed with one integrative approach. Predecessors of this integrative approach have been suggested earlier as key frameworks to unravel complex assembly rules (see refs ¹⁻³) that could not be addressed when focusing only on a single (or limited number of) arbitrary choices. We now state this motivation more clearly in the introduction (Lines 65-71).

R2.2:

Although the workflow is really impacting and well developed I feel that the theoretical framework is weaker. For instance, do you really think that the information provided by the functional diversity facet and the phylogenetic one is directly connected? Or, that the prevalence between one or the other is only a question of spatial scale? If so, what you were saying is that the really relevant information is the phylogenetic although at small scales, the functional one, could be a very efficient alternative. This needs a clearer information.

We do not assume any of these relationships. We investigated productivity-biodiversity relationships for 144 facets, 2 ecosystem types, 3 climate zones, and 3 levels of land use intensity. This wealth of relationships assessed made it impossible to develop specific hypotheses for every single relationship and thus the comparisons made in the third paragraph of the introduction should be interpreted as *tendencies* about what we expect to increase or decrease with the parameters a , δ , and q .

Regarding expectation of functional vs. phylogenetic diversity, we argue that – under certain assumptions – they can be proxies for species' niche overlaps and by this would influence productivity. We felt this is not the place for a full discussion on all assumptions, but wanted to express that, for example, functional diversity can be a good proxy when the measured functional traits are those that are relevant for species environmental requirements, biotic interactions and effects on the ecosystem. The argument for phylogenetic diversity is similar. Finally, functional and phylogenetic diversities can give largely overlapping information if both are good proxies for niche overlap, or in other words, if the traits we consider are relevant for species' niches AND are conserved in the phylogeny, following the framework of Cadotte et al.⁴. However, it can also be that only functional diversity (i.e. traits are not phylogenetically

conserved) or only phylogenetic diversity (i.e. relevant traits are not measured directly but are phylogenetically conserved) are good proxies. In this case FD and PD do not necessarily contain the same information. We now actually tested the amount of shared information between functional and phylogenetic dendrograms and found no significant relationships (see R2.4).

For the question concerning spatial scale: with “scale” in the manuscript we mostly did not refer to spatial scale but to functional/phylogenetic similarity scale. Thus, we did not make any hypothesis about the importance of functional/phylogenetic diversity at different spatial scales. We have now clarified our formulations in the third paragraph of the introduction.

R2.3:

On the other hand, I miss also some more information about the relevance of dominance to understand these relationships. Why dominance? I can imagine that other surrogates of primary productivity could be also very relevant, for instance the total cover.

With dominance in the manuscript, we referred to surface area covered per species, a measure distinguishing whether a species was a tiny specialist sitting in a small corner of the vegetation plot or whether it was prevalent, covering almost the entire surface. Note that we did *not* investigate the relationship between productivity and dominance directly, but that we considered the dominance of individual species to different extents when calculating biodiversity estimates. Dominance was either not considered at all, like for species richness, or quite strongly, as for calculating indices related to Rao’s quadratic entropy. As pointed out by Referee 1, in biodiversity measures considering high dominance weighting (high q), rare species hardly contribute to biodiversity, and biodiversity estimates tend to be highest if dominance is uniformly distributed in the community (i.e., high evenness) (R1.3). We now better introduced the concept of dominance and how we used it (Lines 53-57).

How total cover is related to productivity may be an interesting question. However, total cover is a proxy for green surface rather than for biodiversity, and therefore we feel that this question is not well related to our study design to be discussed here. Furthermore, if productivity is measured by NDVI, linking it to total cover may be circular.

R2.4:

I have also many difficulties with the reduced number of functional traits you have used. Do you think this number is high enough? Even more, do you think they are critical enough to explain the diversity of vegetation types you found. For instance, SLA and height seems only relevant in light limited communities. In this sense, seed mass is critical for dispersal and establishment, but, do you think there is some kind of limitation for seed dispersal and early establishment in your region of reference? Even more, you are using central values, but don’t you think other terms related to the distribution of these traits in the population could be also of interest? If so, you could include not only the mean value, but, for instance the Standard Deviation per species.

Obviously this is more work, but, at least, you should try to better explain these questions. In this sense I miss also to know if the three considered traits have a phylogenetic signal or not.

This is a fair point. Although the three traits considered here are of high ecological importance since they belong to the leaf economics spectrum⁵, they only allow for a crude estimate of functional diversity, and they may not mirror all essential trade-offs acting along the productivity gradients in the ecosystem types investigated. Given the massive data set at hand, for each trait we needed estimates for at least 1200 species in order to be able to consider it, an amount of information that is simply not available for most other traits. For standard deviation rather means of traits, the problem gets even worse, as at least 3-5 independent measurements per species would be needed.

We now added a sensitivity analysis, using a subset of the observations, where we showed that adding leaf dry matter content (LDMC) and leaf nitrogen content (LNC) hardly affected productivity-biodiversity relationships (Supplementary Fig. 1*) and discuss the findings in our “limitations” paragraph in the discussion (Lines 337-358). Furthermore, we tested the phylogenetic signal in the traits considered using Mantel tests and found no significant correlations (Lines 460-463).

R2.5:

In your vegetation data set, a very different type of plant communities is included. The most shocking differences are related to the fact that you included natural communities with a very low-level of disturbance together with plant communities submitted to regular and intense man-driven management such as cutting or manuring in some semi-natural grasslands. Obviously this implies drastic and dramatic differences, especially in the case of some grasslands. From my perspective this can obscure your results and conclusions. In addition, some of your vegetation groups are reasonably homogeneous, such as forests, but other are very internally heterogeneous. For instance, within grasslands you can find intensively managed vegetation types together with almost pristine and untouched pastures like those of some alpine zones. In my opinion this makes inferring conclusions very complex. As a recommendation I suggest to organize your data set with some estimates of man-driven perturbation levels.

This is a valid concern and a good suggestion. We have addressed the issue with an additional, final part of the analysis, where we grouped grassland observations by three levels of land use intensity (Fig. 6* and supplementary Fig. 6*, Lines 251-262). The classification assumed that during the growing season, the amount of variation in NDVI observations that is left after correcting for the seasonal signal, is directly linked to the frequency of land use practices such as mowing, grazing, or manuring – an assumption that has proven to be useful⁶.

R2.6:

Although of less importance I also miss some more information on the theoretical background of the so-called Theory of Coexistence. What are you talking about? You only briefly introduced the idea of filtering and limiting similarity, but as you know patterns of functional clustering or

overdispersion can be obtained through very different mechanisms. If so, how could you assign an observed functional pattern and structure to a specific mechanism. This is critical if you want insert your discussion in this theoretical background.

What we tried advocate at the example of modern coexistence theory is that by testing predictions of one theory along several dimensions (i.e., biodiversity facets in different ecosystem types) we have a more powerful way of assessing its relevance than by just testing its predictions for one bivariate relationship. However, we absolutely agree with Referee 2 that with our approach we cannot infer with confidence whether there is causality. We therefore concluded that “modern coexistence theory offers a consistent explanation for the observed diversity of productivity-diversity relationships”, rather than claiming that it is *causing* the patterns. We now describe the idea of modern coexistence theory and the point we aim to make with it more clearly (Lines 316-336).

R2.7:

An additional problem is related to the fact that forests in French Alps are very poor at least in the tree guild, so observed dominance in them is very intense. Comparison with other vegetation types is then, very complex with vegetation plots structured in an almost binomial structure, very dominated (forests) to lowly dominated (the rest).

High relative dominance of few forest species would lead to low values in biodiversity facets with high dominance weighting. Biodiversity estimates across all ecosystem types, where forests are associated with rather high productivity (Fig. 4*) do not show the pattern proposed by Referee 2 (biodiversity mostly increases with higher productivity & more forests). As pointed out in *R2.3*: be aware that we did not directly compare average dominance in forests with average dominance in grasslands, but that we varied the importance of species dominance in biodiversity estimates.

R2.8:

We need some more information about the binning of the data set and the selection of 18 bins of increasing NDVI.

We now justified the number of bins chosen better in the methods (lines 502-506).

R2.9:

At the end of the first paragraph of the Discussion section there is a sentence that says nothing, “In the following paragraphs we compare the patterns with the recent literature.....”.

We removed the sentence.

I have some difficulties with the dichotomy you posed on the table in relation to the connection between niche separation and competitive ability. As you suggested, these two ideas are

orthogonal but in my opinion, niche separation is closely related with your, as species, ability to compete. Probably I am wrong, but a more detailed explanation will be acknowledged.

As we understand it, coexistence theory assumes that niche separation and competitive ability are orthogonal axes⁷. As mentioned in *R2.6*, we did not aim to further develop this theory here but rather assessed whether it was consistent with our multidimensional patterns (see *R2.6*).

R2.10:

A very minor concern is that I do think the annual plant communities should be eliminated from the final data set. The problem is that the NDVI of this plots should be restricted only to the season in which these plants were monitored. IN this sense, minor questions, I miss some more information about the way in which numerical classifications are done. The only thing it is commented in the text is that “.....derived hierarchical clusters” There is no information of the algorithm. The rule of thumb for considering an acceptable model of $R^2 < 0.15$ is too generalist. Could you include some more information to support this point?

Annual plant communities were not included in our data, which had a focus on perennial grassland and forest communities. We now mentioned that we used the unweighted pair group method with arithmetic mean (UPGMA) algorithm for average linkage clustering (Lines 439-442) and cited a reference that highlights its relevance for building functional trees, and justified our choice of $R^2 < 0.15$ as a threshold to interpret curve types (Lines 561-564).

Detailed response to comments of Referee 3

R3.1:

The paper examines the relationships between species richness, trait diversity or phylogenetic diversity and productivity. Analyses of species richness vs productivity and a multi-faceted measure of trait-phylo-diversity vs productivity are conducted. The team finds that these relationships are different and species richness-productivity relationships are not enough to really understand biodiversity ecosystem function relationships. I suppose the ultimate claim her (which summarizes the first paragraph of the paper and the paper generally) is not really that novel. It reminds me of the work by Cadotte et al. (PNAS and PLoS One papers) from about a decade ago and even Tilman et al. from two decades ago (Tilman et al. 1997 Science). That is, we have realized for quite some time that there is a lot more to B-EF research than simple plots of species richness vs some variable.

We think that two research avenues should be distinguished here: B-EF research and the study of biodiversity changes along the productivity gradient. The papers of Cadotte and Tilman mentioned by Referee 3 do include FD and PD, but they refer to B-EF research, i.e., they are based on local manipulation experiments testing the hypothesis that higher biodiversity leads to higher productivity. They typically include a limited number of species, making the assessment of FD and PD comparably easy. Conversely, we investigated how biodiversity changes along the productivity gradient. Our approach is non-invasive and assumes a much broader

perspective than B-EF experiments. It compares very diverse communities and arguably addresses “one of the most relevant and long standing question in Ecology“ (Referee 2). Notable recent papers along this research avenue are refs. ⁸⁻¹⁰. Working with dominance information, traits, and phylogenies at these scales is very data-hungry (1200 species, 11000 community plots) and to our knowledge has not been done yet. So, we do believe that our analysis stands out from the previous literature already through its sheer scale, left aside our 3D approach to biodiversity facets (*R3.2*). We have introduced these two research avenues now in the second paragraph of the introduction and mention that more biodiversity facets may have been considered in B-EF research. In addition, changing our terminology following the suggestion of Referee 1 (*R1.1*) should help to keep the two research avenues apart now.

R3.2:

The novel twist of this paper may be the 3-d mixing of traits-phylogeny-dominance. It blends Cadotte's framework using the α parameter, pagel's tree transformations in a somewhat interesting way. Thus, from a methods perspective, this makes some kind of advance on how we may compute something new. However, clarifying how this new measurement leads us to new biological insights is quite unclear.

- (1) It allows us to address productivity-biodiversity relationships independent of arbitrarily chosen biodiversity indices and thus allows drawing more general conclusions.
- (2) It provides a better test-bed for ecological theory as it allows assessing predictions along several dimensions rather than only along a bivariate relationship.

We now advocated these points more clearly in introduction and concluding paragraph (see also *R2.1* and *R2.6*).

R3.3:

This brings me to the crux of my critique. The approach largely explores parameters space in a large dataset. The hypotheses set forth at the end of the introduction that are linked to a particular pattern are certainly not the only hypotheses that could give the pattern described. Thus, at the end we have a statement that parameter values 1,2 and 3 fit the data the best, but we really don't know why and strong statements supporting one model (e.g. stress gradients) or another (e.g. limiting similarity) are not reasonable.

The aim of this study clearly goes beyond doing some simple tests of bivariate relationships and identifying the biodiversity parameterization that yields the highest R^2 . The objective of the study is to show the relevance of relationships between productivity and biodiversity should be investigated in concert across a representative range of meaningful facets for a rigorous assessment of ecological theory, motivating future developments of theory to allow for predictions along the different facets, as is the case for modern coexistence theory.

While we had mentioned these ideas in discussion and concluding paragraph, we did not do so in the introduction of the initial submission of the manuscript, where we rather started out with a

set of simple, traditional hypotheses. This likely has made the communication of these points inefficient. We have now rewritten large parts of the introduction to put these points forward, which should make aim and novelty of our study more evident from the very beginning.

R3.4:

In sum, the main conclusion being drawn is not all that different from key take home messages from Tilman and Cadotte in the past and the new analytical approach does not provide new biological insights. I consider the work solid, but unlikely to move the field forward.

In sum, we did not conduct manipulation experiments on local communities but worked with a “massive” (Referee 2) observational data set covering large spatial scales at which working with dominance, traits and phylogenies is novel. We do not claim that our approach is the only one or the best one to integrate these different components of biodiversity, but we do believe that investigating relationships between productivity and biodiversity dimensions in concert across different bioclimatic zones, ecosystem types, and land use intensity levels allows for deeper biological insights and will move the field forward.

Bibliography

1. Chalmandrier, L., Münkemüller, T., Lavergne, S. & Thuiller, W. Effects of species' similarity and dominance on the functional and phylogenetic structure of a plant meta-community. *Ecology* **96**, 143–153 (2015).
2. Graham, C. H., Storch, D. & Machac, A. Phylogenetic scale in ecology and evolution. *Glob. Ecol. Biogeogr.* **27**, 175–187 (2018).
3. Machac, A., Graham, C. H. & Storch, D. Ecological controls of mammalian diversification vary with phylogenetic scale. *Glob. Ecol. Biogeogr.* **27**, 32–46 (2018).
4. Cadotte, M., Albert, C. H. & Walker, S. C. The ecology of differences: assessing community assembly with trait and evolutionary distances. *Ecol. Lett.* **16**, 1234–1244 (2013).
5. Wright, I. J. *et al.* The worldwide leaf economics spectrum. *Nature* **428**, 821–827 (2004).
6. Griffiths, P., Nendel, C., Pickert, J. & Hostert, P. Towards national-scale characterization of grassland use intensity from integrated Sentinel-2 and Landsat time series. *Remote Sens. Environ.* 111124 (2019). doi:10.1016/j.rse.2019.03.017
7. Mayfield, M. M. & Levine, J. M. Opposing effects of competitive exclusion on the phylogenetic structure of communities. *Ecol. Lett.* **13**, 1085–1093 (2010).
8. Adler, P. B. *et al.* Productivity Is a Poor Predictor of Plant Species Richness. *Science (80-.)*. **333**, 1750–1753 (2011).
9. Grace, J. B. *et al.* Integrative modelling reveals mechanisms linking productivity and plant species richness. *Nature* **529**, 390–393 (2016).
10. Fraser, L. H. *et al.* Worldwide evidence of a unimodal relationship between productivity and plant species richness. *Science (80-.)*. **349**, 302–305 (2015).

Reviewers' Comments:

Reviewer #1:

Remarks to the Author:

I have read the new version of this manuscript and the responses to the review comments. I'm fully satisfied with both and support publication of this important paper. The new version is now taking full advantage of the rich data to compare grassland with forest and group these into different strata according to land-use and environmental conditions. The most interesting result are the different responses of species richness vs. functional/phylogenetic diversity to productivity across the entire data set and in forests. The meaning of this finding will have to be more deeply considered in future biodiversity research.

I have only two remaining questions:

1) is it correct to say forests have "higher NDVI(productivity)" than grasslands or is this a place where a caveat should be added that NDVI may not so well reflect productivity across ecosystem types, because it does not see the productivity removed by land use. This also relates to the surprisingly low correlation between land-use intensity and NDVI in Fig. 6.

2) Are the phrases "occupy these niches" and "existing niches" appropriate considering that niches are a property of species but not of the environment?

Reviewer #2:

Remarks to the Author:

Review Brun et al., The productivity-biodiversity relationship varies across diversity dimensions.

I have just read with interest the new version of the original paper. I do think the paper is very valuable and also acknowledge the effort done. It is clear for me that they have done a very fruitful and serious work in an attempt to meet the requirements for publication in an outstanding journal. They also have done a very serious work for responding all the concerns arisen by the three reviewers.

Although I am very conscious that to work in an observational context always produce difficulties in order to isolate the real causal picture, the work done by the authors is really impacting and well justified. This means that their effort helps to surpass the limitations of this type of approach and also, I acknowledge the changes introduce to make this flow easier.

In any case some difficulties remain. For instance, the abstract is definitively very poor and no very informative. To say that something has been rarely done for justifying a paper is not very straight forward. There are thousands of things rarely conducted and without any interest. They should improve this explanation knowing that the abstract is critical and probably the part of the paper more frequently read. This is also a problem that remain in the introduction and also in the discussion. I am very conscious they have done a very nice work in their review, but centered in methodological questions and leaving difficulties in the theoretical background. For instance, I miss clearer expectations, perhaps in the format of a table, of what they think that the productivity – diversity relationships should vary between diversity dimensions, and for what. Their conclusion which literally is "These different relationships for biodiversity metrics and ecosystem types may be understood by linking species similarity to niches and competitive ability as advocated by modern coexistence theory" is very vague. How do their results move the plant community ecology science forward? Why should community ecologists evaluate the three diversity dimensions in parallel? What is the definitive take home message? As presented in the conclusions and especially in the abstract the paper seems a

methodological proposal for dealing with vegetation data sets in the future.

I also find these difficulties in the title which gives no information of the main result.

In my opinion, the discrepancy between the excellent data set and statistical analyses and the theoretical context is important which makes the final impact of the result elusive. For instance, authors use the concept of species similarity/dissimilarity for explaining these relationships. This can be done when the phylogenetic and functional diversities are considered but this is not possible in the case of the taxonomic diversity where these similarities only can be evaluated at the level of the realized assemblage. This is important because the theoretical context for the productivity biodiversity relationships was developed with the taxonomic diversity. This implies a novel framework which needs explanation.

In definitive I feel that the paper maintains some discrepancies between the methodology and the presented conceptual achievements which makes the final result a little questionable.

I do think that working a little bit more in the theoretical framework and re-writing the abstract the paper could become very interesting for an ample audience.

Reviewer #3:

Remarks to the Author:

I reviewed this work previously (previous Reviewer 3). My main critiques at that stage were that the link between productivity and biodiversity (using multiple facets) was a well-traversed path. I also was concerned with whether this is essentially a methods paper that relies heavily on previous methods (e.g. Cadotte's work) where the outcomes rely on a specific parameter (his "a" parameter) that is, in most cases, arbitrarily chosen or fit. Ultimately, I definitely feel the work is useful and publishable, but I find it to be a blend of another paper on a large pile of diversity-productivity papers and a methods paper. This is not a bad thing, but it suggests to me it probably isn't the major breakthrough required for a journal such as this.

The responses to my 2 main critiques were:

1) the data are "massive" and I mischaracterized their work as experimental manipulations B-EF research whereas they are correlating biodiversity measures across a productivity gradient.

2) The method provides a "test bed" for ecological theory for, among other things, so-called modern coexistence theory.

My comments are largely the same. Here they are in response:

1.1) A "massive" dataset does not mean it is worthy of being published in a top journal. This is fallacy. Sure, this does happen, but we really need to move past the idea that compiling a large dataset is how one gets published in a major journal. We are in the era of big data. Data sets on the scale presented here are not that unusual or surprising. If anything, a dataset on the country scale is considered small at this point. If we end up reading it and saying, ok we didn't learn much more, but it really was a large data set. I'm not confident that is a major advance. It appears the authors feel otherwise.

1.2) It is certainly fair not to characterize the present work as experimental B-EF work. It is not. However, the method analyzed is derived from such work and now being applied to a productivity gradient. That was my only point. Beyond that, correlating of biodiversity with productivity gradients is also a large literature. A large fraction of biodiversity gradient research correlates some variable

(e.g. NDVI) with biodiversity as a measurement or proxy of productivity. So, a major response that this work isn't experimental and therefore unique/novel rings hollow. I don't see it.

2) The main interesting aspect of this paper is the adaptation of Cadotte's method and blending it with phylogenetic information. I like it. It should be published. I would expect this to be something found in *Methods in Ecology and Evolution* (i.e. a top level methods journal). I just don't see this as enough of an advance in our understanding of biological systems to merit publication in the present journal. It is a blending of existing methods. Again, that's fine. As for this being a "test bed" of ecological theory. It, like other approaches, can be used to quantify correlate variables to see if expected relationships occur. That is. Again, that is fine, but we should be careful not to overstate the importance of things. This is not going to solve modern coexistence theory debates or other debates more directly linked to the paper (i.e. biodiversity along productivity gradients). It helps, but let's not overdo it.

Thus, I am afraid my opinion remains the same. The biological knowledge gained is incremental and the methodological approach is interesting (though itself an incremental advance done by blending existing metrics). The argument that the work should be published primarily because the data are massive does not resonate.

Detailed response to comments of Referee 1

I have read the new version of this manuscript and the responses to the review comments. I'm fully satisfied with both and support publication of this important paper. The new version is now taking full advantage of the rich data to compare grassland with forest and group these into different strata according to land-use and environmental conditions. The most interesting result are the different responses of species richness vs. functional/phylogenetic diversity to productivity across the entire data set and in forests. The meaning of this finding will have to be more deeply considered in future biodiversity research.

I have only two remaining questions:

- 1) is it correct to say forests have "higher NDVI(productivity)" than grasslands or is this a place where a caveat should be added that NDVI may not so well reflect productivity across ecosystem types, because it does not see the productivity removed by land use. This also relates to the surprisingly low correlation between land-use intensity and NDVI in Fig. 6.

This is a fair point. Since NDVI is a measurable quantity, the correctness of this part of the statement is clear. However, while NDVI generally works as a proxy for productivity, we agree that its limitations may be more severe when comparing between ecosystem types rather than within ecosystem types. We have removed the interpretation "(productivity)" in the results and highlight the limitation of NDVI when comparing between ecosystem types in the caveats section of the discussion.

- 2) Are the phrases "occupy these niches" and "existing niches" appropriate considering that niches are a property of species but not of the environment?

We agree with Referee 1 that, by themselves, the statements may appear confusing as the term ecological niche is often associated with the Hutchinsonian definition that indeed describes niches as a property of the environment. However, in biodiversity science the relatively common terms 'niche space' and 'niche dimensionality' describe the structure of the habitat and the diversity of ecological strategies it can support. 'Niche' in this context follows more the original definition of Grinnell¹. We have now replaced 'niche' with 'niche space' where it was appropriate and we state in the introduction that we use the term 'niche' in the Grinnellian sense.

Detailed response to comments of Referee 2

Review Brun et al., The productivity-biodiversity relationship varies across diversity dimensions.

I have just read with interest the new version of the original paper. I do think the paper is very valuable and also acknowledge the effort done. It is clear for me that they have done a very fruitful and serious work in an attempt to meet the requirements for publication in an outstanding journal. They also have done a very serious work for responding all the concerns arisen by the three reviewers.

Although I am very conscious that to work in an observational context always produce difficulties in order to isolate the real causal picture, the work done by the authors is really impacting and well justified. This means that their effort helps to surpass the limitations of this type of approach and also, I acknowledge the changes introduced to make this flow easier. In any case some difficulties remain. For instance, the abstract is definitively very poor and no

very informative. To say that something has been rarely done for justifying a paper is not very straight forward. There are thousands of things rarely conducted and without any interest. They should improve this explanation knowing that the abstract is critical and probably the part of the paper more frequently read. This is also a problem that remain in the introduction and also in the discussion.

We have a different view here. We do not justify our work by saying it has been “rarely done”. Rather, we state that we address the lasting, major challenge of understanding the different processes that drive the dramatic biodiversity changes along the productivity gradient. We have now clarified this as far as possible within the 150 words constraint.

I am very conscious they have done a very nice work in their review, but centered in methodological questions and leaving difficulties in the theoretical background. For instance, I miss clearer expectations, perhaps in the format of a table, of what they think that the productivity – diversity relationships should vary between diversity dimensions, and for what.

We have already carefully illustrated our expectations in Figure 1b and 1c. We now more explicitly refer to these panels.

Their conclusion which literally is “These different relationships for biodiversity metrics and ecosystem types may be understood by linking species similarity to niches and competitive ability as advocated by modern coexistence theory” is very vague. How do their results move the plant community ecology science forward? Why should community ecologists evaluate the three diversity dimensions in parallel? What is the definitive take home message? As presented in the conclusions and especially in the abstract the paper seems a methodological proposal for dealing with vegetation data sets in the future.

We have replaced this statement with a more concrete, ecological conclusion.

I also find these difficulties in the title which gives no information of the main result.

The complexity of our analysis makes it impossible to explicitly describe our main result in a short and catchy title. We believe that the proposed title is a sound compromise.

In my opinion, the discrepancy between the excellent data set and statistical analyses and the theoretical context is important which makes the final impact of the result elusive. For instance, authors use the concept of species similarity/dissimilarity for explaining these relationships. This can be done when the phylogenetic and functional diversities are considered but this is not possible in the case of the taxonomic diversity where these similarities only can be evaluated at the level of the realized assemblage. This is important because the theoretical context for the productivity biodiversity relationships was developed with the taxonomic diversity. This implies a novel framework which needs explanation.

In definitive I feel that the paper maintains some discrepancies between the methodology and the presented conceptual achievements which makes the final result a little questionable.

I do think that working a little bit more in the theoretical framework and re-writing the abstract the paper could become very interesting for an ample audience.

We have rewritten the abstract according to the suggestions of Referee 2 and of the responsible editor.

Detailed response to comments of Referee 3

I reviewed this work previously (previous Reviewer 3). My main critiques at that stage were that the link between productivity and biodiversity (using multiple facets) was a well-traversed path. I also was concerned with whether this is essentially a methods paper that relies heavily on previous methods (e.g. Cadotte's work) where the outcomes rely on a specific parameter (his "a" parameter) that is, in most cases, arbitrarily chosen or fit. Ultimately, I definitely feel the work is useful and publishable, but I find it to be a blend of another paper on a large pile of diversity-productivity papers and a methods paper. This is not a bad thing, but it suggests to me it probably isn't the major breakthrough required for a journal such as this.

The responses to my 2 main critiques were:

- 1) the data are "massive" and I mischaracterized their work as experimental manipulations B-EF research whereas they are correlating biodiversity measures across a productivity gradient.
- 2) The method provides a "test bed" for ecological theory for, among other things, so-called modern coexistence theory.

My comments are largely the same. Here they are in response:

1.1) A "massive" dataset does not mean it is worthy of being published in a top journal. This is fallacy. Sure, this does happen, but we really need to move past the idea that compiling a large dataset is how one gets published in a major journal. We are in the era of big data. Data sets on the scale presented here are not that unusual or surprising. If anything, a dataset on the country scale is considered small at this point. If we end up reading it and saying, ok we didn't learn much more, but it really was a large data set. I'm not confident that is a major advance. It appears the authors feel otherwise.

Indeed, we do. Of course, the size of the data set cannot be the only justification. But we are convinced that for this type of analyses it is particularly important to have massive data available in order to enable analyses of community observations along the entire productivity gradient^{2,3}. Here, we do this to our knowledge for the first time in combination with trait, phylogenetic, and dominance information, allowing us to study much more relationships. And these relationships are not just flat but they show strong and contrasting patterns allowing for insight in underlying processes in unprecedented depth.

1.2) It is certainly fair not to characterize the present work as experimental B-EF work. It is not. However, the method analyzed is derived from such work and now being applied to a productivity gradient. That was my only point. Beyond that, correlating of biodiversity with productivity gradients is also a large literature. A large fraction of biodiversity gradient research correlates some variable (e.g. NDVI) with biodiversity as a measurement or proxy of productivity. So, a major response that this work isn't experimental and therefore unique/novel rings hollow. I don't see it.

We consider it as a strength that this work is not experimental, because it allows an undistorted view on natural processes rather than drawing conclusions from artificially manipulating communities along short gradients only. There certainly have been comparable observational studies but as mentioned previously, to our knowledge they, so far, they have worked with species richness and thus had a very narrow view on biodiversity. Here, we expand this scope

to include functional and phylogenetic diversity dimensions and we do so in a integrative and exhaustive approach.

2) The main interesting aspect of this paper is the adaptation of Cadotte's method and blending it with phylogenetic information. I like it. It should be published. I would expect this to be something found in *Methods in Ecology and Evolution* (i.e. a top level methods journal). I just don't see this as enough of an advance in our understanding of biological systems to merit publication in the present journal. It is a blending of existing methods. Again, that's fine. As for this being a "test bed" of ecological theory. It, like other approaches, can be used to quantify correlate variables to see if expected relationships occur. That is. Again, that is fine, but we should be careful not to overstate the importance of things. This is not going to solve modern coexistence theory debates or other debates more directly linked to the paper (i.e. biodiversity along productivity gradients). It helps, but let's not overdo it.

Thus, I am afraid my opinion remains the same. The biological knowledge gained is incremental and the methodological approach is interesting (though itself an incremental advance done by blending existing metrics). The argument that the work should be published primarily because the data are massive does not resonate.

We thank Referee 3 for acknowledging that the work is methodologically sound. As for the relevance of the findings, we will probably not reach an agreement. In our view, the main finding is that sites with high productivity typically have reduced species diversity, but high functional and phylogenetic diversity, and that this potentially is a direct result of creating additional niche space. We consider this novel and a direct result of the method developed here.

References

1. Grinnell, J. The Niche-Relationships of the California Thrasher. *Auk* **34**, 427–433 (1917).
2. Adler, P. B. *et al.* Productivity Is a Poor Predictor of Plant Species Richness. *Science (80-.)*. **333**, 1750–1753 (2011).
3. Fraser, L. H. *et al.* Worldwide evidence of a unimodal relationship between productivity and plant species richness. *Science (80-.)*. **349**, 302–305 (2015).